# Processing and Advancements in the Development of Thermal Barrier Coatings: A Review

**Amrinder Mehta [1], Hitesh Vasudev [2,3,]*, Sharanjit Singh [1], Chander Prakash [2,3], Kuldeep K. Saxena [4,]*, Emanoil Linul [5,]*, Dharam Buddhi [6] and Jinyang Xu [7,]***

1   Department of Mechanical Engineering, DAV University, Jalandhar 144012, India
2   School of Mechanical Engineering, Lovely Professional University, Phagwara 144411, India
3   Division of Research and Development, Lovely Professional University, Phagwara 144411, India
4   Department of Mechanical Engineering, GLA University, Mathura 281406, India
5   Department of Mechanics and Strength of Materials, Politehnica University Timisoara, 1 Mihai Viteazu Avenue, 300 222 Timisoara, Romania
6   Division of Research & Innovation, Uttaranchal University, Dehradun 248007, India
7   State Key Laboratory of Mechanical System and Vibration, School of Mechanical Engineering, Shanghai Jiao Tong University, Shanghai 200240, China
*   Correspondence: hiteshvasudev@yahoo.in (H.V.); saxena0081@gmail.com (K.K.S.); emanoil.linul@upt.ro (E.L.); xujinyang@sjtu.edu.cn (J.X.)

**Abstract:** Thermal barrier coating is critical for thermal insulation technology, making the underlying base metal capable of operating at a melting temperature of 1150 °C. By increasing the temperature of incoming gases, engineers can improve the thermal and mechanical performance of gas turbine blades and the piston cylinder arrangement. Recent developments in the field of thermal barrier coatings (TBCs) have made this material suitable for use in a variety of fields, including the aerospace and diesel engine industries. Changes in the turbine blade microstructure brought on by its operating environment determine how long and reliable it will be. In addition, the effectiveness of multi-layer, composite and functionally graded coatings depends heavily on the deposition procedures used to create them. This research aims to clarify the connection between workplace conditions, coating morphology and application methods. This article presents a high-level overview of the many coating processes and design procedures employed for TBCs to enhance the coating's surface quality. To that end, this review is primarily concerned with the cultivation, processing and characteristics of engineered TBCs that have aided in the creation of specialized coatings for use in industrial settings.

**Keywords:** thermal barrier coating; yttria stabilized zirconia; air plasma spray; solution precursor plasma spray; functionally graded coatings; multi-layered coatings; thermal conductivity; thermally grown oxide

## 1. Introduction

This study's main objective is to reduce carbon dioxide emissions from the atmosphere and meet the population's energy demand without harming the climate. A gas turbine can operate at a higher temperature and achieve maximum work output, with minimum energy loss. The advanced gas turbine engine will work in modern times using renewable or non-renewable energy resources for industrial applications, such as producing electricity and operating for the aerospace engine. With the aid of thermal barrier coating in this field, the market area is expected to reach USD 30.7 billion by 2025. Various researchers have found that gas turbine engine modification improves the superior turbine blade efficiency [1]. Technological improvement in gas turbines plays a vital role in filling rising energy requirements and reducing the carbon dioxide emission rate. The challenge facing gas turbines is to prevent their working parts from degradation, as the turbine works at a higher temperature and a corrosive environment [2]. The steam turbine engine's work

quality depends on a higher gas inlet temperature and reduced energy loss during turbine blade operation. The combustion chamber's temperature is maintained inside by using the turbine component's insulation with the surface coating process's help, preventing the turbine component from degradation. Sometimes, the working temperature of steam turbine blades is above the range of melting temperature of the base metal, and overcoming this problem is to use a suitable coating method. By efficiently supplying internal air cooling, the substrate material's surface temperature can be reduced up to 300 °C [3]. TBCs have been developed to insulate the substrate material's surface layer and prevent thermal shocks and corrosive environmental conditions during work. The thermal spray process is widely used for coating the gas turbine component because it can coat intricate shapes. TBC can also affect several factors, such as heat flux, heat transfer coefficients, coating thickness, and thermal conductivity, to define the substrate's magnitude of temperature decrease. These constant efforts to increase a gas turbine engine's efficiency have resulted in working temperatures to over 1300 °C, which requires thicker TBCs with their chemistry development, including the newly cooling system [4]. If the topcoat thickness increases, the substrate coating components' surface temperature decreases from 4–9 °C per 25 μm and avoids surface coat failure under higher-temperature oxidation conditions [5]. The thermal stability of the turbine components at higher temperature thermal shock resistance and wear resistance using conventional or nanostructured coating methods is also helpful for coating the complex shape components for gas turbines or diesel engines. Increased steam turbine engines' efficiency eventually provides practical assistance in aircraft performance and fuel efficiency and supports them in carrying increased payloads. The gas turbine engine produces the maximum amount of mechanical work output for low wastage of thermal energy in land-based applications. It helps generate electricity in thermal power stations. Figure 1 shows that the compressor sucked the air from the environment, then compressed that at a higher pressure; after this, it delivers this air into the combustion chamber, where it helps for burning the all-fuel particles that help for generating the higher amount of heat energy inside the combustion chamber and reduce the wastage of unburned fuel particles. The turbine inlet temperature is similar to the combustor outlet temperature and turbine blades are used to convert the heat into mechanical energy by revolving them into high speed [6].

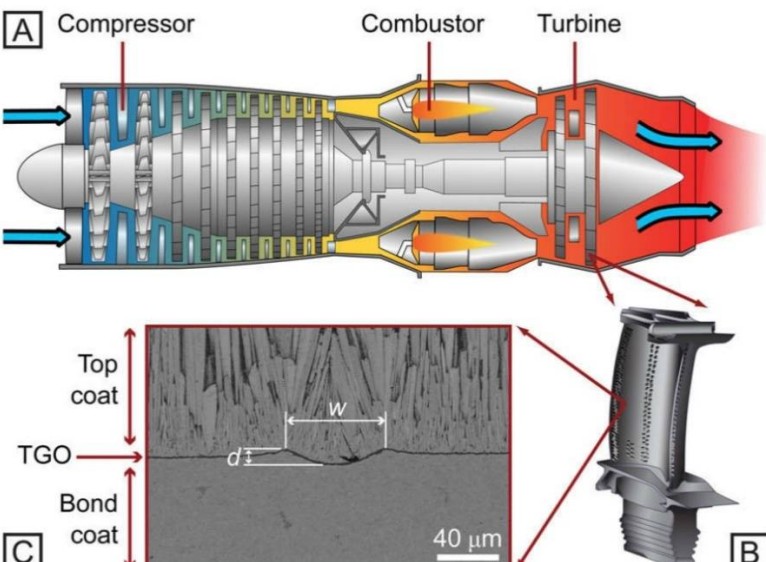

**Figure 1.** (**A**) Schematic diagram of a gas turbine, (**B**) Blade of the turbine and (**C**) cross-section TBC system [6].

The thermal efficiency (η) of a gas turbine following an ideal Carnot cycle [7]:

$$\eta = 1 - (T_{exit}/T_{inlet}) \tag{1}$$

where, $T_{exit}$ = represent the exit temperature of the gas, $T_{inlet}$ = represent the inlet temperature of the gas.

Equation (1) shows that increasing the gas turbine's inlet temperature can improve its thermal efficiency [7]. Raising the turbine's inlet temperature sets an immediate requirement on the materials working in the gas turbine's hot regions, where the temperatures approximate the range of 1500 °C [8]. Here, the temperature is higher than the melting point of the most reliable working materials in gas turbines now, such as nickel-based superalloys. When rising the temperature of hot gas inside the turbine inlet, it is more important to control the temperature and protect the turbine component from degradation. Hence, it is feasible to use outside insulation of the surface material and an internal cooling system for turbine components. This internal cooling system is used to restrict the volume of air and control the heating temperature without decreasing the gas turbine's thermal efficiency. When a quantity of air higher than 8% is utilized for cooling purposes for the turbine, the main benefit of this cooling system is to regulate the working temperature and increase the mechanical and thermal efficiency of turbine blades [8]. Coating quality advancements can improve work performance and reliability [9]. The researchers' foremost challenge issues in surface coating are how to improve the thermal stability of the ceramic powder material at a higher working temperature to provide a better insulation property for the substrate metal [10]. The various approaches and synthesis mechanisms can be used for materials processing in other applications [11]. The details of papers published on thermal barrier coatings have been fetched from scopus.com (accessed on 23 August 2022) and presented in Figure 2. Researchers are focusing on the challenges associated with thermal barrier coatings, and advancements are still going on in this field.

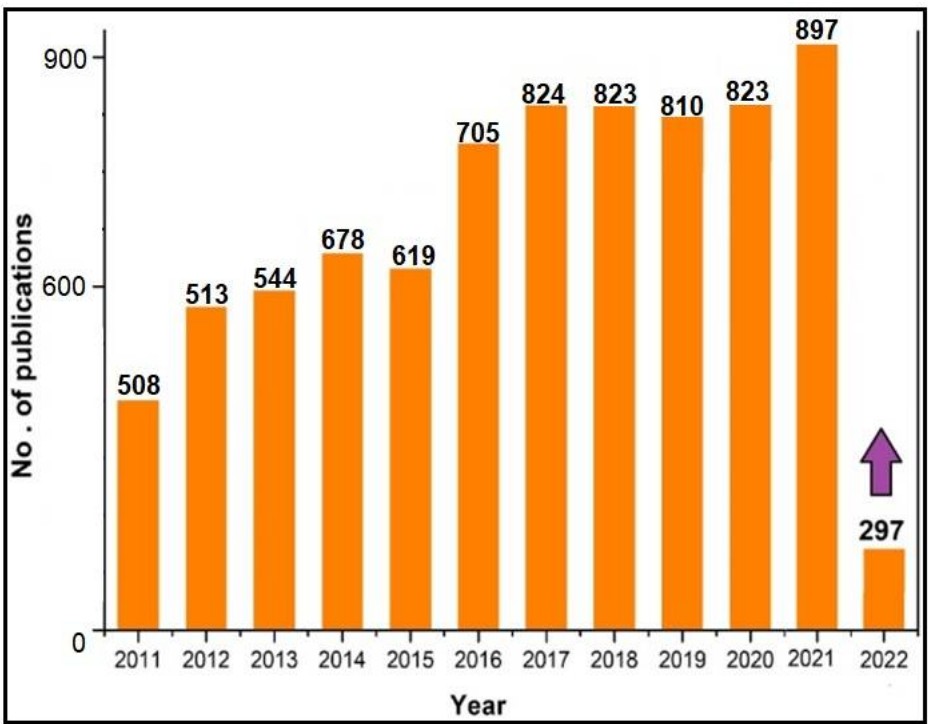

**Figure 2.** Year-wise data published on thermal barrier coatings.

TBCs are used to insulate gas turbine components that are subjected to excessive temperatures. Since low thermal conductivity allows these coatings to act as thermal barriers, metal surface temperatures decrease. Lower metal temperatures are given greater

component durability by reducing creep stress and fatigue, while also reducing oxidation and corrosion rates. Advances are going on in the field of high-temperature applications. Materials that can sustain high temperatures need to be optimized. Thermal barrier coatings need optimization in terms of resisting high temperature oxidation, erosion and hot corrosion. This paper emphasizes the recent techniques used in the development of thermal barrier coatings and their performance has been analyzed. The authors have incorporated the need for this review article into the revised manuscript.

## 2. Thermal Barrier Coating System

TBCs are coated processes used for intricate surface material systems that are useful for gas turbine engines implemented in hot regions. This multi-layered coated surface has a formation where the top coating layer comprises a lower thermal conductivity ceramic and thermal insulation from hot burning gases [12]. The schematic is shown in Figure 3. At maximum engine power, the modern turbine inlet temperature (TIT) of state-of-the-art jet plane turbines reaches a value of approximately 1500 °C. The component temperature level is higher than the maximum operating temperature of YSZ TBCs, which is around 1300 °C. Underneath the topcoat is a bond coat that contributes a vital role for bonding effect between the ceramic material and base metal for protecting turbine components against oxidation and corrosion effect at high temperatures and crossover working environment.

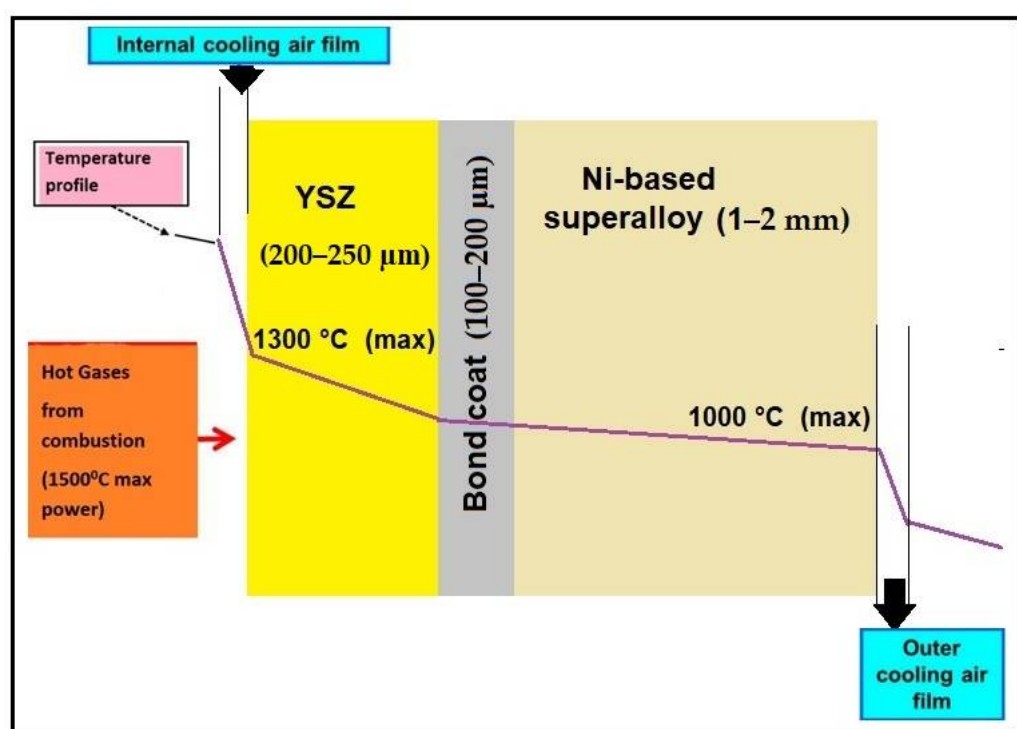

**Figure 3.** Schematic of the Thermal Barrier coating (TBC) part in thermal gradient and air–cooling.

Figure 4 shows the SEM images of a polished cross-section of traditional as-sprayed and nanostructured TBCs, which constituted the YSZ topcoat and the NiCoCrAlY bond coat collected by the APS process on the superalloy substrate metal. After the solidification process was completed for the partially molten and completely molten powder grains, the conventional and nanostructured YSZ covered layered form. Micropores can be seen in the cross-section of the YSZ coating in both layers, as well as on the coating's outer surface. Nevertheless, in the nanostructured coating produced from very fine grains, the inhomogeneities and micropores are frequently decreased, most likely due to the compactness of the nanostructure [13].

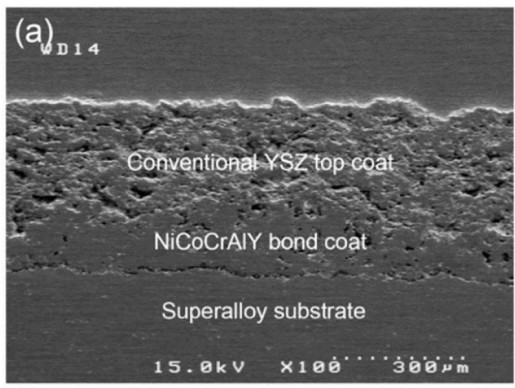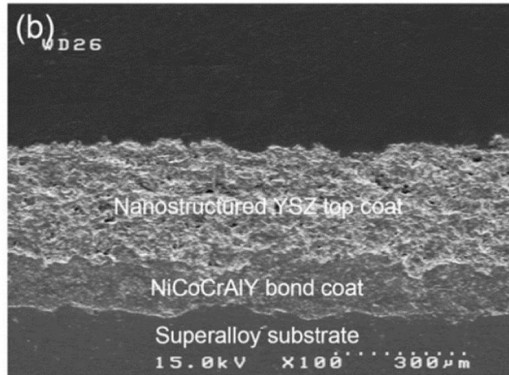

**Figure 4.** (**a**) FESEM polished cross-section micrographs of As-sprayed TBC with traditional or Conventional YSZ coating and (**b**) Nanostructure YSZ coating [13].

*2.1. Material in a TBC System*

2.1.1. TopCoat

The thermal spray coatings' upper zone contributes heat resistance against the burning gases into a gas turbine engine's combustion chamber. The durability and topcoat performance are increased by providing an internal cooling system that keeps the base metal's heating temperature lower, which is why it does not decrease its load-bearing ability. This heating temperature can decrease from 150–250 °C in the upper layer topcoat, depending upon the coated layer's thickness and the type of ceramic grain size [13]. The ceramic powder materials have been investigated for insulation purposes for gas turbine blades by a thermal spray coating process used to improve the thermal stability of the topcoat materials. Yttria-stabilized zirconia (YSZ) powder was mainly used for industrial application because it provided better thermal stability. YSZ ceramic powder particles need to have many required characteristics for the working heat limit, such as increased melting point temperature, reduced thermal conductivity, and improved phase stability [14].

The lower coefficient of thermal expansion is a significant factor in the improved life of coated surface material controlled by proper coating methods. A single ceramic material can be challenging to fulfil all of the coated ceramic materials' conditions, using a mixture of two or more ceramic materials to overcome these situations. In addition to yttria-stabilized zirconia in the range of 6–8 wt.%, this gives the topcoat powder grains better performance because of its comparatively lower density, lower thermal conductivity and higher concentration for point defects [15]. Many researchers want to develop the YSZ coating structure's excellent quality, which improves the components' durability and reliability at a higher working temperature [16]. In the opposite direction, YSZ already has a few drawbacks for coating utilization, such as reduced working temperature less than 1200 °C, susceptibility to thermal corrosion, impressive atmospheric sediments, and stimulated thermally grown oxide generation large diffusivity of ionic oxygen in ceramics based on $ZrO_2$ [17].

Hence, it is essential to improve the coating material's work performance and provide a more stable coating microstructure. To prevent the substrate metal from deteriorating at higher temperatures and corrosive environmental conditions. The 8YSZ/graphite powder material is used for high-temperature applications. It has good insulation properties that protect substrate metals by reducing the heat transfer rate between the coating and substrate surfaces [18]. At present, various researchers have worked to produce agglomerated nanostructurednanostructured8 wt.% YSZ powder grains that can be used for the thermal spray coating process. The size range of conventional powder grains varies from 10 to 100 μm, but on the other hand, the size range of nanostructured powder grains varies from 40 to 50 nm. In the thermal spraying process YSZ, nano-clusters use feedstock to control the supply of powder grains during the working process. The nano-clusters agglomerate powder grains during spray drying, which can help obtain micro-sized particles for TBC coating [19]. The coating quality relates to the complete molten for the powder grains

before being deposited on the base material in the thermal spray process. The chemical reaction technique plays a significant role in manufacturing nanostructured powder grains and makes them capable of surface coating. The spray-drying method is an essential and valuable technique for agglomerating nanostructured powder grains. This technique helps protect them from overheating before depositing onto the substrate material [20]. The various kinds of powder grains material utilized in combining multi-layered topcoat structured ceramics could be a successful option.

### 2.1.2. Bond Coat

Many scientists have improved the bond strength obtained from nanostructured powder grains. As a bond coat below a ceramic topcoat in TBC systems, MCrAlY has been commonly employed. By practising standard C633-01, the shift from the bond coat hardness is proposed to occur guided by this adhesive force when that breakdown occurs from the bond-coated base metal in this interface. That cohesive force fracture is entirely within this coated layer [21]. Consequently, this bonding of the power inside the associated splitting is stronger than that without splitting. It is well understood that combining governs the adhesive strength of the situation within sprinkling materials, including the base material. The YSZ-based coating material manages residual tension from the base material's coated form [22]. This kind of residual stress is made from a couple of parts in that sprayed method, such as quenched stress that produces reduction for the various splats and cooled pressure generated by the unmatched heat extension between that deposited on the base material while all collectively cooled later deposited [23]. One of the promising ultra-high-temperature applicant metals for a material bond coat is the intermetallic composite NiAl, which has a higher molten point (1638 °C), lower density (5.9 g/cm$^3$), relatively greater elastic modulus (240 GPa), and excellent oxidation protection at 1200 °C [24]. On the other hand, the functional utilization of NiAl under a higher-temperature environment is restricted by its brittleness, low adhesion strength within this oxidation film, and base metal. This diffusion process occurs between the Ni and Al atoms because a higher temperature operation toward the model also limits the powder grain characterization of NiAl [25,26]. After performing the experiments, it was observed that the more outstanding durability of bonding from the nanostructured coating was determined by the greater internal toughness achieved [27,28]. As this conventional coated approach watched some inner face inside this powder material, some kept fully melted into the spray blast, and this base metal showed minor cracks. As this nanostructured coated, some interfaces inside this powder grain were partially melted during this spraying process. Adherents in the coating occurred in this base metal. Thus, interface cracks within the nanostructured-coated structure are prevented. The firm-adherent compact nanostructure increases the bond strength.

### 2.2. TBC Processing Methods

As shown in Figure 5, it offers a specific classification of processing techniques and the necessary process employed to deposition TBCs on the substrate metal, which has been mentioned in the following paragraphs. In the field of thermal spray, thermal plasmas are employed in atmospheric plasma spray (APS), suspension plasma spraying (SPS), solution precursor plasma spray (SPPS), Plasma Spraying –Physical Vapour Deposition (PS-PVD) vacuum plasma spray (VPS), low-pressure plasma spray (LPPS), and controlled atmospheric plasma spray (CAPS) methods.

The following sections extensively discuss atmospheric plasma spray (APS) because it is a standard coating method that can model typical thermal spray processes. Because of the relative amount of momentum supplied to the in-flight particles, plasma jet flow alters the trajectories of feedstock [29–31].

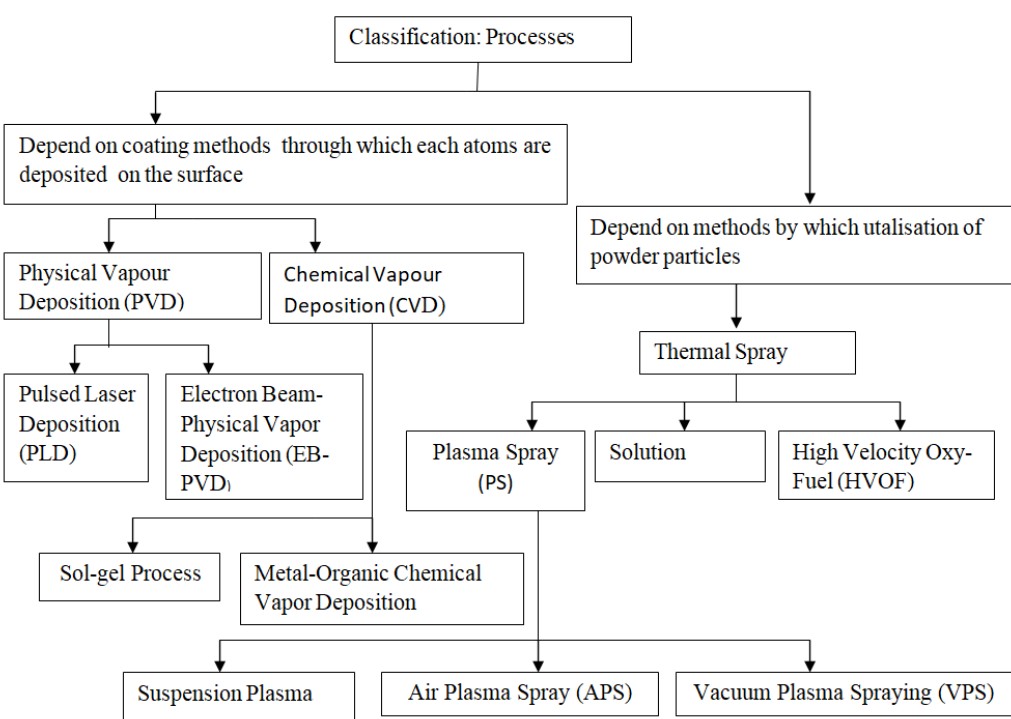

**Figure 5.** Wide classification of the coating process and methods used for material deposition.

2.2.1. Air Plasma Spray (APS)

This TBC material-deposited method typically defines this type of part coating to the spray jet, such as atmospheric plasma spraying (APS). In common, APS coated appear to have a horizontal splatted micro-structure with inter-splatted borders approximately equal for this top-coated and bond-coated interfacial [32]. In this method, powdered material grains are injected in a thermal spray jet of higher heating range and melted within this jet of spray, which works for stimulation into the base metal where melted droplets spread that create splits efficiently quenched. The coated surface thickness can be increased by raising the number of successful splats deposited on the substrate metal. This coated structure quality mainly depends on the process parameters used for thermal sprays, such as the current, voltage, the flow rate of that gas, and the stand-off distance between the touch gun and substrate material during surface coating. The feedstock plays a vital role in supplying a sufficient amount of powder through the spray gun to obtain a successful coating structure. The feeder input parameters used to control this supply of powder grains include the temperature and size of the powder grains. The coated microstructure splats' morphology depends on the angle of contact used to spray the powder grains to the substrate material [33]. Some usually cracked cross-sections for this powder spraying coated surface by APS show films for the splatted simultaneously with inter-laminar holes and round holes [34]. The coating's porosity is structured depending upon the substrate's surface temperature being reduced by increasing heat temperature and improving internal splat attachment, resulting in improved coating characteristics [35].

Figure 6 shows the powder's morphology used for deposition in the substrate metal by the APS method in the existing work explained. The mean size was 40–70 μm for all globular agglomerated powder grains.

Figure 7 displays the FE-SEM images of the APS method-based individual-layered YSZ cross-sectional view, YSZ overlay of $Al_2O_3$, and YSZ coating overlay of $CeO_2$. Besides, $Al_2O_3$ and $CeO_2$ coatings are also considered crack-free and well adhered to the coating of the YSZ.

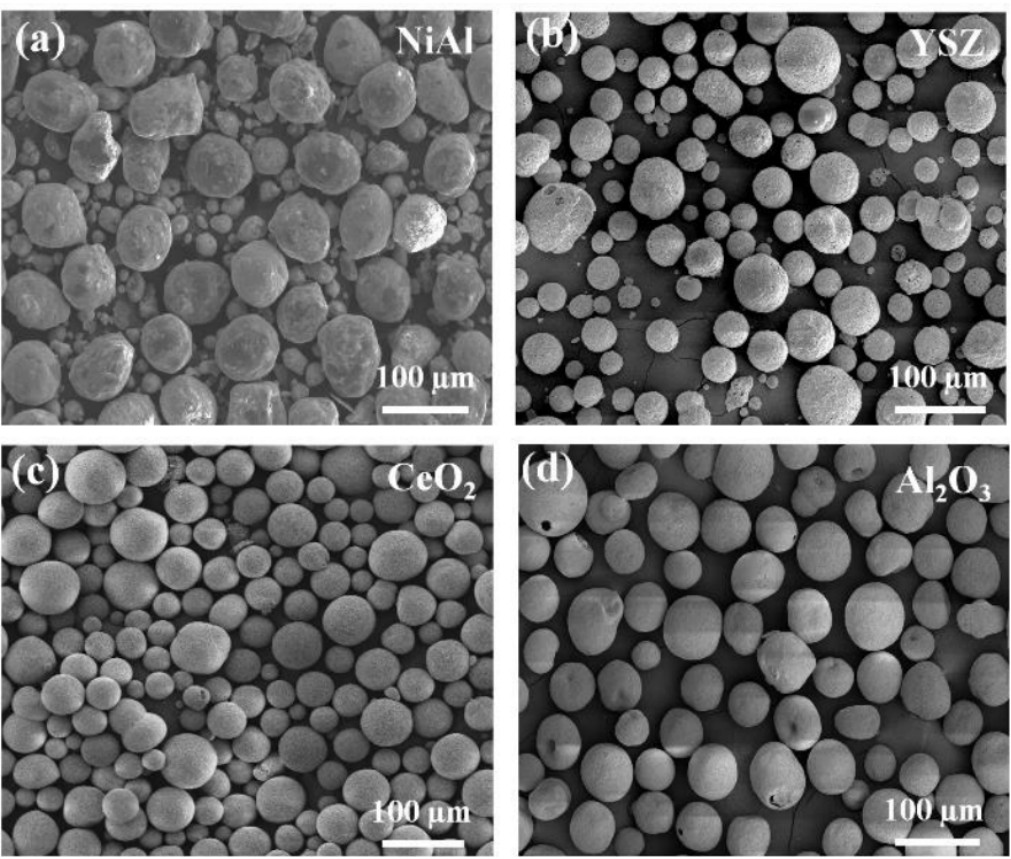

**Figure 6.** FE-SEM images show the morphology of the (**a**) NiAl, (**b**) YSZ, (**c**) CeO$_2$ and (**d**) Al$_2$O$_3$ feedstock powder.

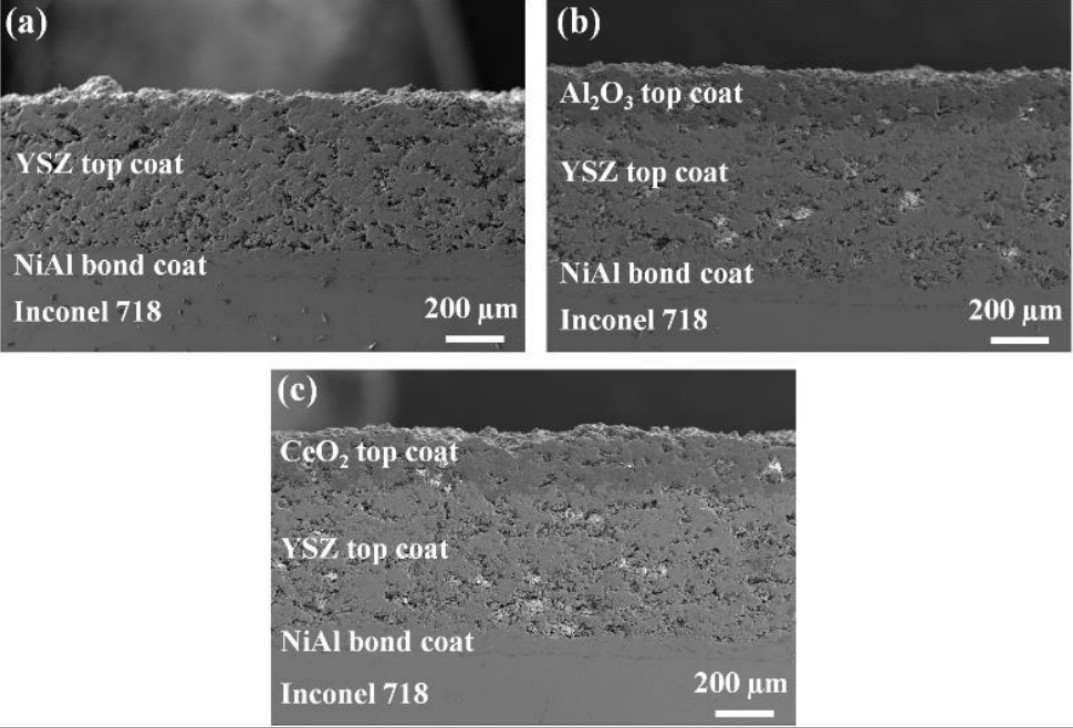

**Figure 7.** Low magnification FE-SEM images show the plasma-sprayed cross-sectional surface (**a**) NiAl/YSZ, (**b**) NiAl/YSZ/Al$_2$O$_3$ and (**c**) NiAl/YSZ/Al$_2$O$_3$/CeO$_2$ [36].

As shown in Figure 8, all three layers contained a bimodal microstructure consisting of an entirely melted section joined collectively to create a solid structure. The sponge-shaped microstructure portion is formed near the molten zone. APS coatings' common characteristics, intermittently partially melted powder grains, micro-level cracks, and porosity play a significant role in the coated structure's thermal oxidation and corrosion resistance behavior. As diffusion paths for molten salts, these partially melted power grains and micro-cracks work by enhancing the corrosion reaction between the dissolved salts and the ceramic powder [36].

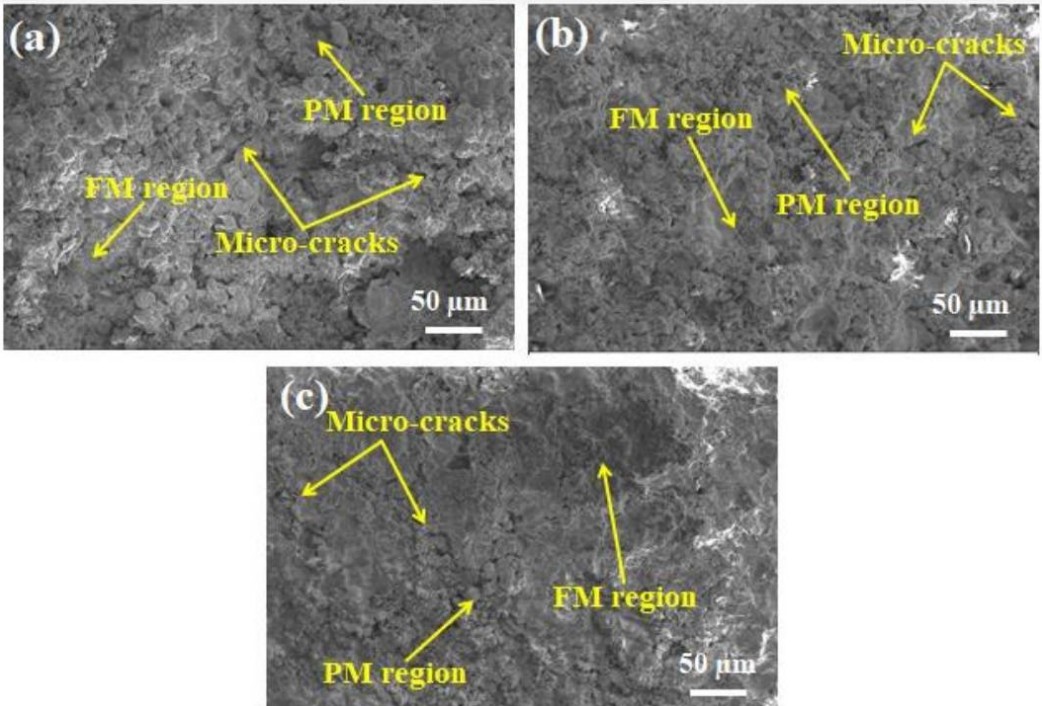

**Figure 8.** FE-SEM micrograph images of (**a**) NiAl/YSZ, (**b**) NiAl/YSZ/Al$_2$O$_3$ and (**c**) NiAl/YSZ/Al$_2$O$_3$/CeO$_2$, based coating structure exhibiting the micro-cracks, partially melted (PM), fully melted (FM) region [36].

Thermally grown oxide (TGO) mainly occurs between the top and bond coating structures because oxidation occurs during work at higher temperatures. The oxide layer growth mainly produced the maximum amount of stress inside the coating structure interface, which reduced the components' service life. Experimental findings revealed that YSZ coating with Al$_2$O$_3$ as an overlay improved traditional YSZ's hot corrosion protection by partly alleviating melted salt infiltration. However, by completely clearing the molten salt's penetration in the YSZ, the YSZ layer with CeO$_2$ as the overlay film showed more effective corrosion protection than traditional TBCs [36]. Young's modulus is the most significant factor in TBCs that influences the thermal stress distribution in the coating structure's upper surface and improves the components' work performance. Because of the porous microstructure, the upper surface's macro-elastic properties are usually significantly less than this value for dense YSZ [37] (see Table 1). The relationship between the various parameters is shown in Figure 9. The failure of the bonded structure in cracks appears near the splats region because of expanding numbers of thermal cycles during the experiment work and increasing the strain value, which affects the reduced macro-elastic strength [38]. To determine the modulus of elasticity for the coated structure with depth-sensitive indentation, the methods were used, and stability depended upon the load applied for nanoindentation. With the heat treatment method's help, the modulus of elasticity can be improved by heating the coated surface at 1100 °C [39].

**Table 1.** The plasma spraying parametersfor the top coat's first stage.

| Substrate Material | Powder Feed Rate (g/min) | Current (A) | Voltage (V) | Power (kW) | Stand-Off Distance (mm) |
|---|---|---|---|---|---|
| stainless steel [34] | 20–40 | 600 | 66–68 | - | 100 |
| IN-738LC [35] | 25 | 600 | 60 | 36 | 150 |
| IN-738LC [40] | 38 | 600 | - | 64 | 100 |
| Amdry 995C [37] | 60 | 600 | 65 | - | 120 |
| 204B-NS [37] | 40 | 600 | 65 | - | 130 |
| stainless steel [38] | 30 | 600 | 63 | 50 | 100 |
| stainless steel [38] | 35 | 600 | 67 | 42 | 120 |
| Inconel 718 [39] | 35–40 | 600 | 60 | 35 | 130 |

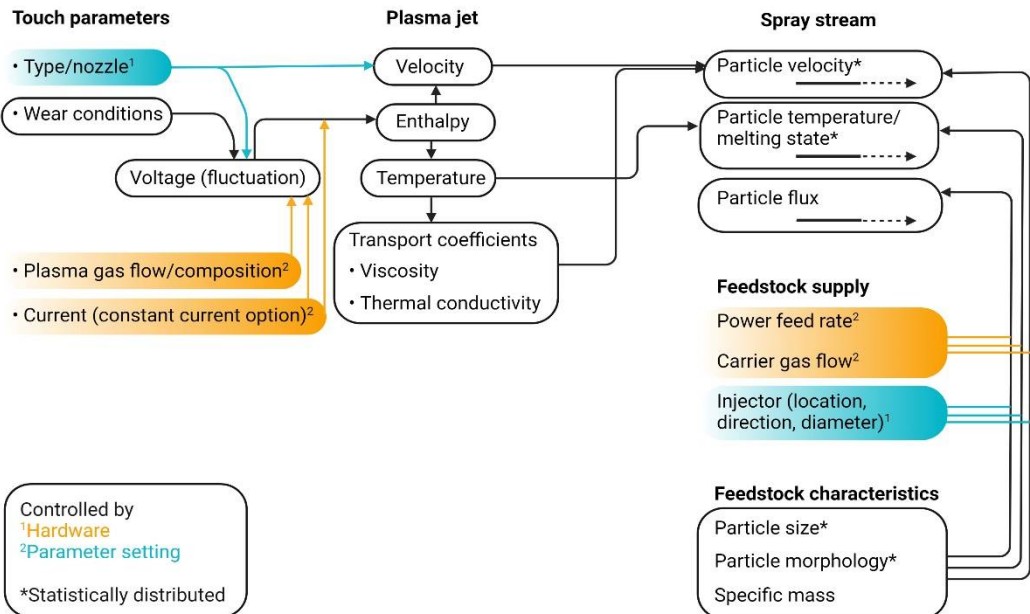

**Figure 9.** The plasma spraying parametersfor the top coat's second stage.

As shown in Figure 10, the H$_2$ or Ar can be used as carrier gases in APS coating. These gases entered from the electrode hole that heated up and disintegrated them through the arc to form plasma. The nozzle can control the jet of plasma in this process by regulating the speed and flow rate of the burning gases. These burning gases are mainly used as carrier gases in this method. They mix into agglomerated nano- or micro-sized diameter powder grains that help melt them before being deposited on the substrate material. Many process parameters have been used in the APS method. However, some of the most significant parameters must be affected by the surface's standardized coated properties and improve the microstructure quality.

Some necessary input parameters that must be controlled for better microstructure from the coated surface, such as the pressure of gas arc feed rate of powder grains, diameters of powder particles, the flow rate of carrier gas, angle of injection, the distance of spray, smoothness of the coated surface and base metal or metal substrate surface temperature before coating start [41] (see Table 1). Several research types have confirmed that the optimum parameters in this process play a significant role in APS. It controls phase structure and thermal durability, improves microstructure, enhances corrosion protection, and improves physical properties, such as the coating's hardness and toughness.

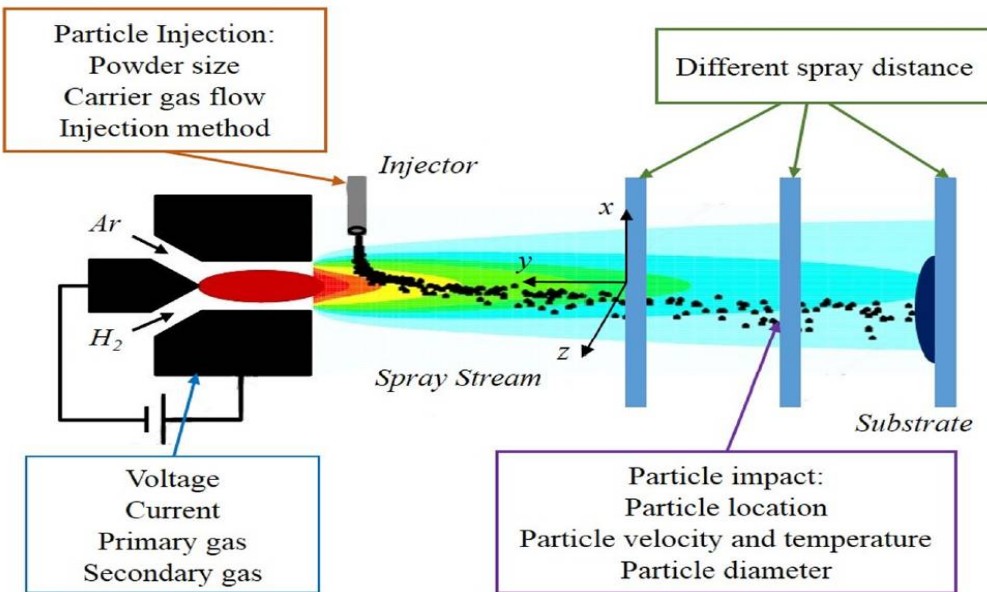

**Figure 10.** Themost important controller parameters used in APS [41].

### 2.2.2. Functionally Graded Coatings

Traditional coated methods have faced significant problems under working conditions. This coated structure usually fails earlier when presented under the thermal shock and erosion test due to this causing the spallation effect produced in a material that can start deteriorated coated structured from the interface between top and bond coat. In functionally graded (FG)coatings, multiple phases are produced and given a regular face structure. The ceramic topcoat faces a higher operating temperature under this coating system and sustains this heat under the combustion chamber, while this also functions as heat insulation [42]. The coating must be done by depositing gradient material at a controlled amount to substrate metal during the process. The gradient material allows for the proper distribution of the stress generated around the coated surface and causes a decrease in strength. The bond coat strength has improved by reducing the thermal expansion gap between the top coated film and base metal after coating. When the higher amount of coated material is deposited through the spray drawing process and the diameter of traditional powder in microns used for coating, then the maximum chances for the spallation at the coating layer's outer surface are then given. Some authors have produced nanostructured powder grains that troubleshoot this material spallation difficulty using nanostructured powder structured in YSZ TBCs material with a higher thermal expansion coefficient and lesser thermal conductivity overall [43]. Different studies have focused on the effectiveness of various kinds of functionally graded coatings in decreasing heat stress and improving the coated structure's mechanical properties, such as thermal shock resistance and excellent hardness value [44]. Some researchers deposit the $Al_2O_3/YSZ$ ceramic powder onto substrate materials and analyze its mechanical and thermal properties. The oxygen entrance is restricted by applying a thin surface coating of $Al_2O_3$ ceramic material and providing substrate metal thermal insulation. Spallation has formed inside the $Al_2O_3$ ceramic thick-coated structure while working at a higher temperature than can change the ceramic material phase from $\gamma$-$Al_2O_3$ to $\alpha$-$Al_2O_3$. The coating material's premature failure occurs when extra residual stress is generated inside the coating structure due to thermal oxidation [45]. This study will design a zirconium base coating material and examine fracture toughness using a theoretical modeling technique. This coating was done through the functionally graded process. A unique parameter is similar to the stress-intensity factor introduced to determine the toughening effect transform toughness equal to stress. The design methodology helps select optimal coating parameters that prevent the coated structure's premature failure and improve the fracture toughness material [46].

Figure 11 represents every layer's purpose inside the functional zones of this functionally graded coating (FGC) technique. In hot corrosion conditions, the stimulation decomposed for the $Sm_2Zr_2O_7 + 8YSZ$ way into the form of APS for the double ceramic layer (DCL) and functionally graded system (FGS) modes was examined. This investigation was to distinguish surface methods in a liquid sodium sulfate $Na_2SO_4$ environment that occur throughout the corrosion of FGC based on samarium zirconate and zirconium oxide with ratios 25/75, 50/50, and 75/25. After experimentation, the selective breakdown of samarium zirconate in the 50 8YSZ + 50 $Sm_2Zr_2O_7$ composition that system gave better corrosion resistance compared to others [47].

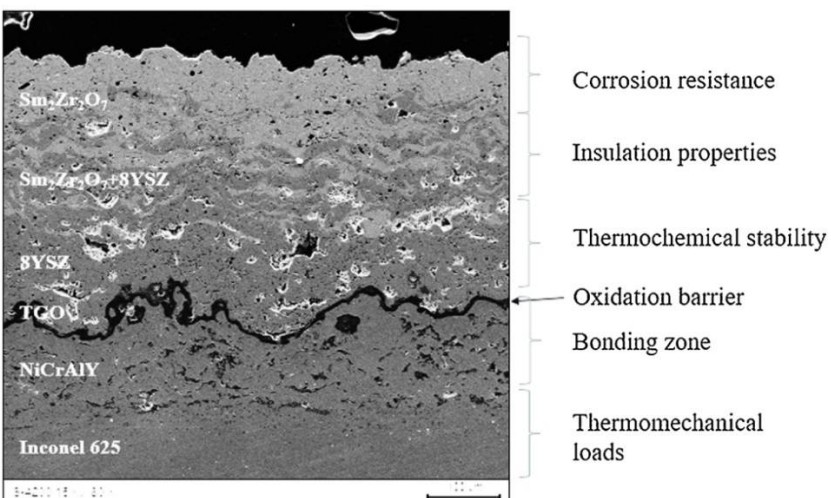

**Figure 11.** FE-SEM images show the functional zones of the FGC technique [47].

The functionally graded coating material with thermal diffusivity/conductivity progressively improved over the functionally graded zones compared to the variation in composition accompanying this coating mechanism; the testing specimens were compared. The thermal oxidation test was conducted at a temperature of 1300 °C for 1 h, and the cooling process was done at 25 °C for 20 min for holding time to completion of one cycle. The composite layers' thermal conductivity and diffusivity improved by increasing the NiCoCrAlY content and temperature, varying from 500 to 1200 °C. This investigation outcome explained that the functionally graded coating material's thermal oxidation stability was five times higher than that of the conventional coating process for equivalent thicknesses. As shown in Figure 12a,b, it represents the thermal diffusivity and conductivity done at a variable temperature (from 25 to 1200 °C) and densities (0, 50, 75 and 100%). Figure 12c illustrates that FGC has improved thermal conductivity five times higher than a conventional coating, and the surface's quality is dependent on each coating layer's porosity level [48].

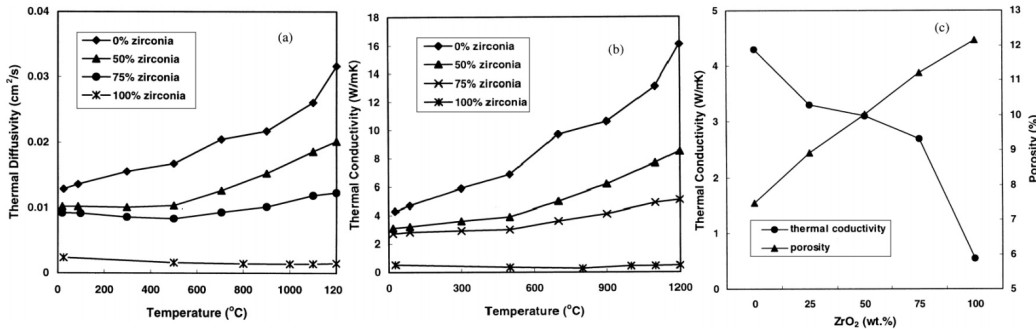

**Figure 12.** The purposes of the FGC mechanism: (**a**) thermal diffusivity, (**b**) thermal conductivity, and (**c**) the correlation between porosity and thermal conductivity in atmospheric temperature [48].

### 2.2.3. Multi-Layered Coatings

Multi-layer topcoat fabrication is an effective method to overcome the common problem and transform it into YSZ-based TBCs. The essential requirement for this multi-layered coated structure is to improve the stability and durability that results are required to increase the work performance. The multi-layer coated microstructure contains different coated films deposited in this coated structure and products, decreasing its thermal radiation effect [49]. A multi-layered idea has been used for the too-reflective coated system, including several stacks that can lower material outside temperatures and increase thermal efficiency [50].

### 2.2.4. Suspension Plasma Spraying (SPS)

The suspension plasma spraying (SPS) system is mainly used for the required high-quality coated microstructure, which has primarily developed from the traditional plasma spraying process. This method can help establish a new composite coated micro-structured quickly onto the substrate material. The coating of structured material features can be improved using agglomerated nanostructured powder grains to reduce porosity [51]. The transport medium shifted from a gas state to a liquid form and observed the development of melted droplets that provided the required impulse [52]. Under this method, high-density state powder grains have been required to apply this surface coating technology, which has provided an uninterrupted supply for the molten droplets during working conditions. The diameter of the selected powder grains varies from 10 to 100 μm [53]. This has remained a complicated duty because this viscosity for the molten feedstock principally affects melted droplet generation inside the jet of spray, and the melting temperature depends on the diameter of the powder grain. This upper surface topography of the coated structure mainly depended on the powder and heating supply rate for melting the powder grains [54]. In this process, some critical liquid state solvents, such as distillation water or de-ionization water, methanol and ethanol, have been employed for this surface coating method and enhance the quality of this coated microstructure.

The time used for evaporation from the liquid state in the jet of spray and surface tension plays a critical role in selecting the solvent [55]. That has found that more time was spent in the distilled water solvent based on suspension droplet-evaporation in the spray plasma jet; in the other case where ethanol was used as a solvent, that droplet took less time as a consequence of faster fragmentation. With two kinds of technology employed in SPS methods, the pressurised vessel and peristaltic pumps carried the sufficient liquid phase solvent and used supply to the spray jet at the required amount. The lower adhesion rate, higher fracture toughness, and higher thermal shock resistance all have structured coating properties that provide better durability or reliability under the required mechanical properties. The ASTM G171-03 standards were used to calculate the surface-coated structure's cohesion rate and provided the other required information related to the coating [55]. The researcher conducted the harness value for the 8YSZ-coated design, which varied from 5 to 14 GPa, by conducting the SPS process experiments under the TBCs. In this process, the powder's selection in coating is based on some important parameters, such as the shape of the crystal structure and the grain particle size, and provides better resistance to the coating surface from degradation [56]. The maximum thermal spray-based organization used 7 wt.% YSZ as a ceramic top-coated structure in TBCs because of the few novel properties given due to higher thermal resistance and enhanced fracture toughness [57]. For the highest efficiency and extended work life for these steam turbine blades, the high-temperature range for that YSZ thermal barrier coating is 1250 °C [58].

Figure 13 shows the system used for the injection and fragmentation of the molten powder grains during deposition on the substrate material. The single and agglomerated powder grains that remain in the suspension are delivered immediately and melted through the plasma jet. Single powder grains provide very tiny lamellae or sub-micrometric globular characteristics that have not flattened [59].

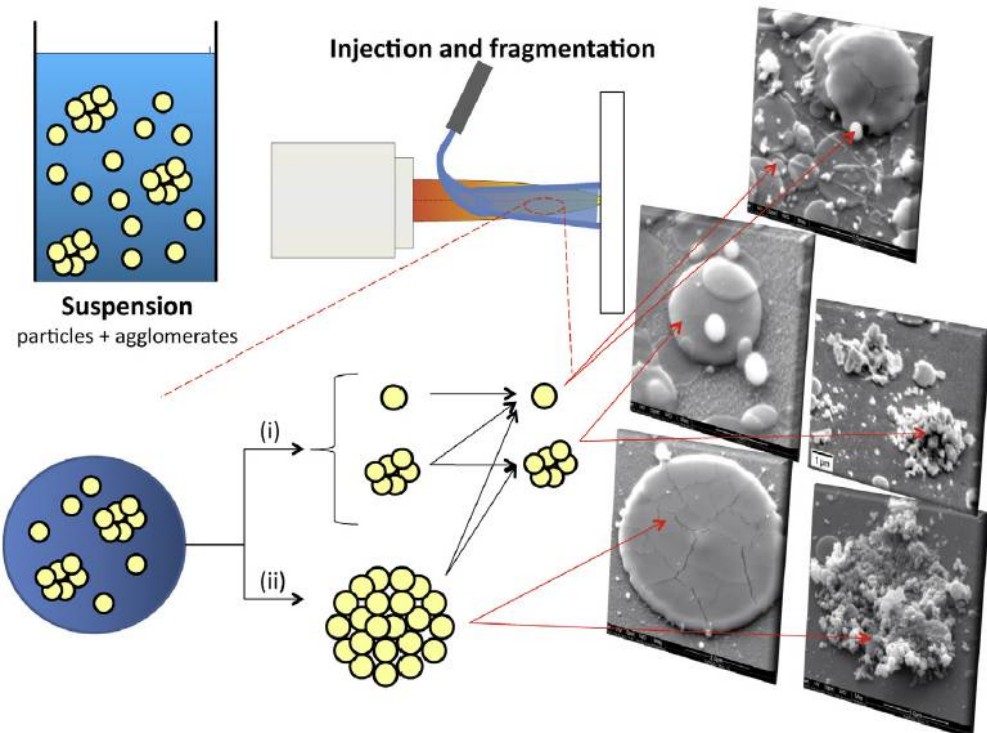

**Figure 13.** Display to deposits system used in the SPS method [59].

According to Van Every K et al. [60], the individual powder grains have to produce a lower amount of inertia to efficiently divert these powder particles. The stagnation of flow on the upper side of the base metal's upper side can decrease the molten material's required velocity, creating a low amount of flattened lamella.

2.2.5. Solution Precursor Plasma Spray (SPPS)

This surface coating technique has been recognized as an appropriate alternative for developing TBC at the lowest cost and has been upgraded for increased thermal cycles and decreased thermal conductivity [61]. The SPPS method, as shown in Figure 14, is very similar to the SPS technique, in which the coating material is supplied into the plasma spray by suspending dense particles into the nanostructured size range. Both procedures use comparable liquid injection methods and have the same problem of evaporating the suspending medium or solvent, which requires a significant amount of energy. The atomized precursor is used as a feedstock to transmit the chemical precursor by injecting this aqueous form solution into the plasma jet. It is also helpful for fast physical reactions inside the plasma region [62]. This method consists of organic or inorganic solvents, compounds of metallic-organic and inorganic salts used to complete the process [63].

The liquid precursor injection technique is similar to the suspension plasma spray technique, but the coating development system is somewhat different. Cracks in SPPS coatings emerge as the deposited semi-pyrolyzed materials contract owing to subsequent in-process heating, as demonstrated in Figure 15b, stress-relieving throughout the thickness. They are not restricted to dense coatings. The fact that the splat sizes are on the 1–5 μm order, more than 10 times less than the average powder-sprayed splats, is another unique aspect of the SPPS process (Figure 15a) [64,65]. There are some essential features of the SPPS-coated method that can be developed as an evenly ordered porosity effect and the regular separation of vertical cracked. The orientation of this base metal helps increase thermal stability and enhances the coated surface's bond strength [66].

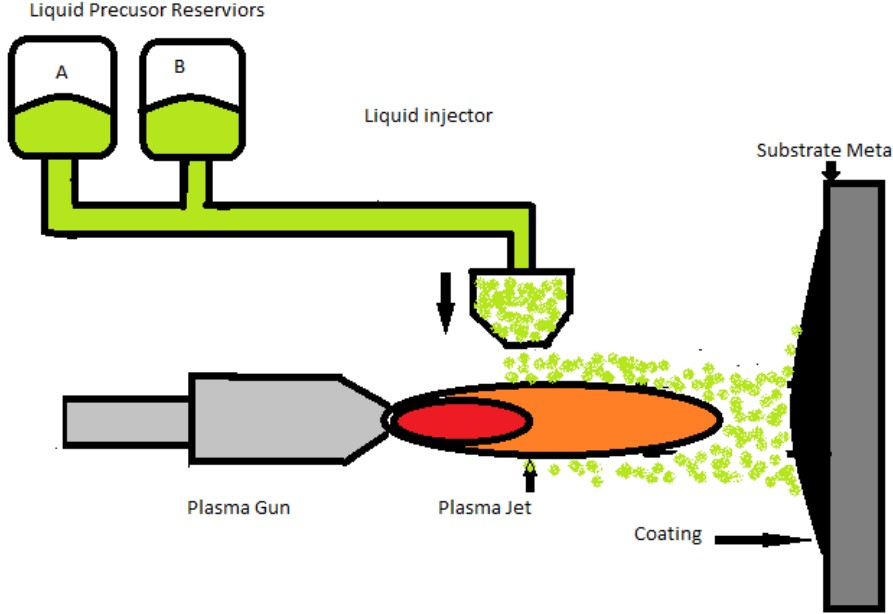

**Figure 14.** Schematic view of a Solution Precursor Plasma Spray system with radial injection.

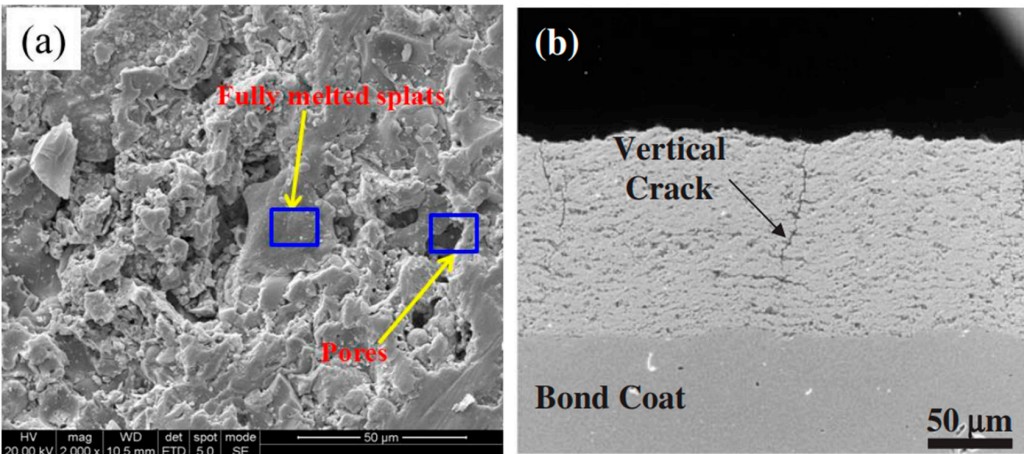

**Figure 15.** The SPPS TBCs have unique microstructural characteristics: (**a**) small splat area, and (**b**) stress reduced by introducing vertical cracks inside the coating [64,65].

The non-melted powder grains and extremely fine-sized splatted deposited on the substrate material must be observed in this coating method. Any non-melted powder grains holding a non-pyrolyzed precursor appear to deteriorate and crystallise throughout the thermal shock test. A lesser stand-off distance must be required in the SPPS method than in the traditional coating method and provided superior quality of a surface structure [67]. The organic or inorganic solution was used to dissolve the micro- or nanostructured powder grains and prepare the suspension. The appropriate dispersion has been used to adsorb the outer region to the powder gains combined into this suspension system to control their stability and prevent these agglomerated powder grains. The suspension system gave less viscosity to dissolve the powder grain at the appropriate level. The suspension's rheological properties have been significantly influenced by particle size, powder surface composition, solid loadings, and dispersant rate [68]. Powder grains are synthesized to be helpful for synchronous deposition onto the coating process's substrate material [69]. The change in grain powder symmetry can significantly impact the relative adjustable chemical structure [70]. The jet of plasma has been classified into three zones, depending on temperature variation: the plasma jet core, plasma plume, and plasma fringe. In this

method, the injection mode, such as suspension and solution, has played a significant role that can be affected by the droplets' heated temperature [71]. The injector is used to precisely control the molten droplet's speed before depositing it onto the base metal. Other instruments, such as peristaltic and plunger pumps, can transport liquid material in this system. Having the fluid penetrate inside the plasma jet core is desirable and comparatively simple to accomplish under this axial injection method. The radial injection method has commonly been accepted commercially because of its low deposition cost and better-quality deposition [72]. The extra power required for automatizing the fluid at a slower speed can overcome this problem by employing Ar gas because it has a greater specific mass. The spray gun nozzle can carry two different liquids and produce the atomization effect at the coating [73].

### 2.2.6. Plasma Spraying-Physical Vapor Deposition (PS-PVD)

The injected powder material is melted into liquid droplets and evaporates to form a coating using this method. This coating method is used in plasma spray and is designed to minimize the oxidation effect throughout deposition phases to the base material's ceramic materials. The plasma spraying physical vapor deposition (PS-PVD) process is further subdivided into two categories: low-pressure plasma spraying (LPPS) and vacuum plasma spraying (VPS). Improvements under advanced vacuum plasma spraying methods, such as plasma spraying physical vapor deposited (PS-PVD), plasma spraying chemical vapor deposited (PS-CVD), and thin-film LPPS, have recently occurred. The gas-phase use for coating and the chamber operating pressure (>100 Pa) have some significant characteristics for this method [74]. Compared to conventional Air Plasma Spray, the PS-PVD is used as an advanced coating method. It can provide the ceramic material's with fast deposit onto the substrate metal. To improve the LPPS coating method's working efficiency by increasing the vacuum pressure from 50 to 200 Pa inside the processing room, the 200 NLPM (Normal Liters Per Minute) required a flow rate and 180 kW power to operate the plasma gun for a spray for plasma gas for the coating process. Thin film technology, which allows for the creation of uniform, homogenous coatings, may produce the thinnest coatings [75]. The protection of the base material from overheat was a major challenge during the coating process and overcame this controlled movement of the plasma gun. Some researchers have developed the bond coat material structure and controlled the plasma gun's power requirement, which depends on the coating material composition and powder particles [75]. The evaporation of the powder grains starts in the PS-PVD process due to the higher temperatures achieved for the plasma plume during the working process. The smoother powder grains provided more benefits for evaporation starting in the plasma plume in the deposition process. The coated surface's microstructure depends on the parameters and the powder grains' diameter during the coating process [76]. The compound of $Ar/He_2$ gases is used to generate the plasma in the PS-PVD process and the YSZ powder grains' feed rate is 80 g/min for obtaining optimum work performance. It is pumped into the plasma gun to guarantee that the powder particles penetrate the plasma core. The nanostructure cluster is made from a columnar structure shape due to the lesser feed rate of powder grains and generation $Ar/He_2$ gases for plasma. The EB-PVD coating method isalso suitable for obtaining a columnar structure shape at a 2 g/min required feed rate of the coating material [76]. The spray parameters can control the type of layer structure, transforming it from a splat-type coating to a porous layer type and eventually to a columnar microstructure.

As shown in Figure 16, the columnar structure requirement is achieved by combining a low powder feed rate, a specific mixture of argon, secondary plasma gases and a considerable spray distance [77].

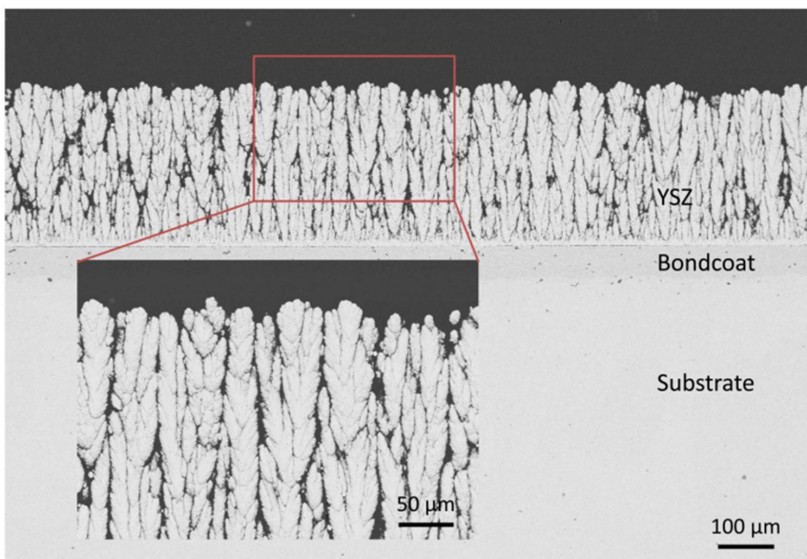

**Figure 16.** Cross-sectional SEM images of a YSZ thermal barrier coating fabricated by PS-PVD [77].

### 3. Failure Mechanism in Thermal Barrier Coatings

The mechanism responsible for the failure of thermal barrier coatings is discussed in this section. The TBCs have to withstand high temperature oxidation, hot carrion and erosion. The rapid oxidation of materials at high temperatures caused by a thin layer of dissolved salt deposits is hot corrosion. The TBCs should have the properties to resist such failures and these have been presented in Figure 17.

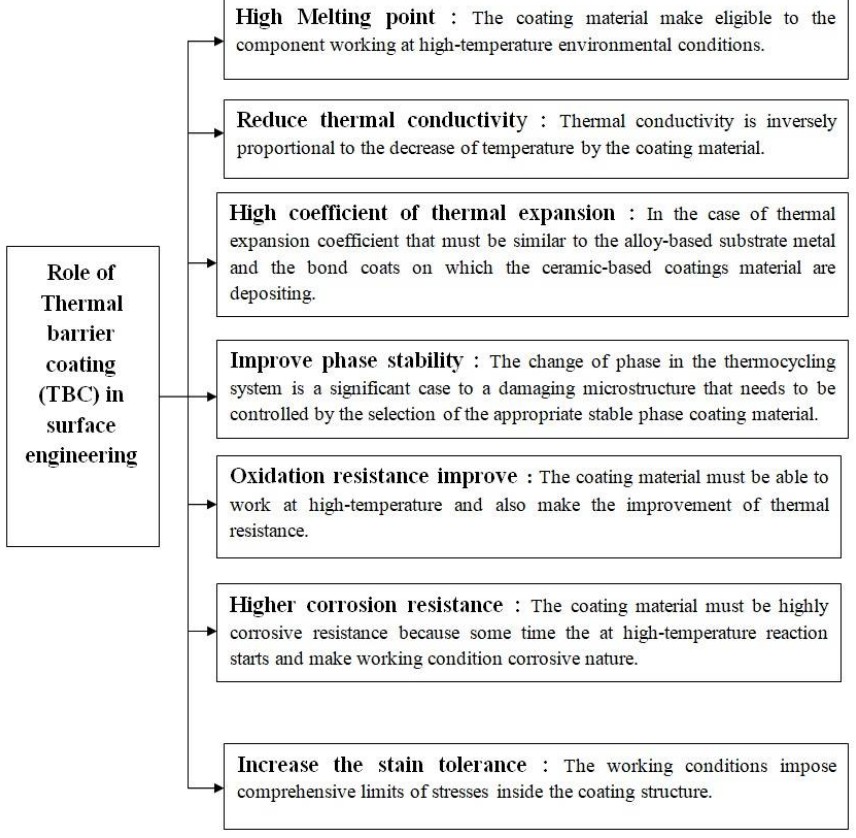

**Figure 17.** The importance and requirement of Thermal Barrier Coating (TBC) in surface engineering applications.

*3.1. Some Challenges and Required Properties in the Selection of Coating Material*

TBCs are coated processes used for intricate surface material systems that are useful for gas turbine engines implemented in hot regions.

TBC coatings are exposed to the harsh environment of the combustion chamber. The following challenges face TBCs: hot corrosion, thermal oxidation performance and hardness values.

3.1.1. Hot Corrosion

The rapid oxidation of materials at high temperatures caused by a thin layer of dissolved salt deposits is hot corrosion [78]. Sulfur is present in lower-grade fuels, where it produces $SO_2$ on combustion and is slightly oxidized to $SO_3$. Sodium chloride (NaCl) combines with water vapor and $SO_3$ at combustion temperature to form sodium sulphate ($Na_2SO_4$), which has a melting point of 884 °C [79].

Low-carbon steel and corten steel are mainly used to manufacture boiler parts in fired coal-based thermal power plants. Superheater tubes, water walls, and economizers are essential parts of the pant. They are exposed to extremely high temperatures, pressures and corrosive environments during the plant's working. This corrosive environment of the original boiler atmosphere contains sulfur, carbon and oxygen, which can cause material degradation and early boiler tube rupture, especially in higher temperature ranges (Figure 18) [80].

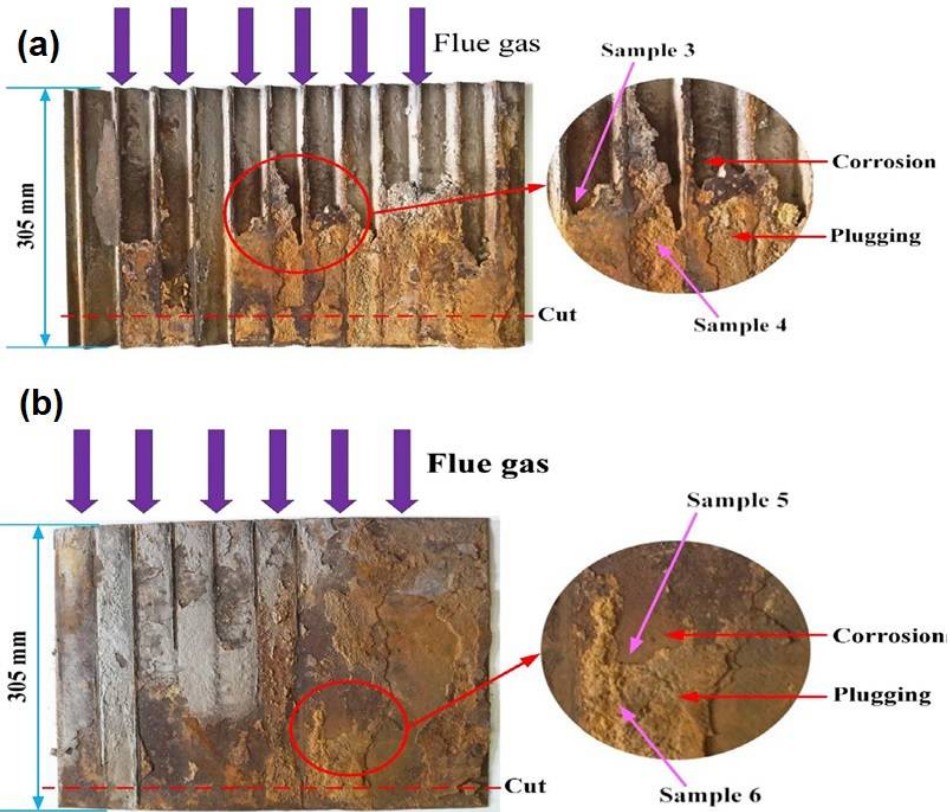

**Figure 18.** Effect of corrosion on (**a**) corrugated plate and (**b**) positioning plate of air preheater (lower portion) [80].

Different Kinds of Hot Corrosion

There are two sorts of attacks when it comes to hot corrosion (Figure 19):

(a) High-temperature hot corrosion (HTHC)
(b) Low-temperature hot corrosion (LTHC)

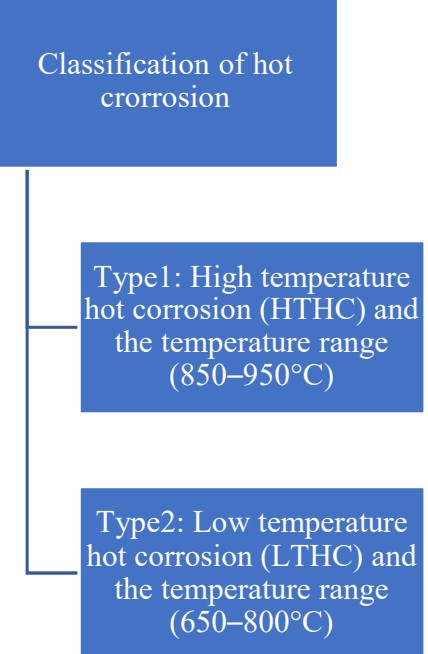

**Figure 19.** Classification of hot corrosion based on temperature.

The coating material and coating method must be selected according to the application of the metallic component [81].

Mechanism of Hot Corrosion and Stages

Numerous authors have proposed several methods to characterize the hot corrosion method [81]. As shown in Figure 20, the hot corrosion of metal alloys occurs in three different stages. The corroded metal alloy part's final failure occurs after the third stage is completed.

**First stage**

In this step, the reaction proceeds at a rate similar to conventional oxidation.

The incubation stage is another name for it.

**Second stage**

In this step, the corrosion is stimulated.

The initiation stage is another name for it.

**Third stage**

In this stage, fast corrosion happens.

The propagation stage is another name of it.

**Figure 20.** Mechanism of hot corrosion and stages.

Preventive Strategies to Combat the Effect of High Temperature

As shown in Figure 21, some preventive techniques have been proposed for excessive material degradation caused by high-temperature corrosive environmental conditions [82].

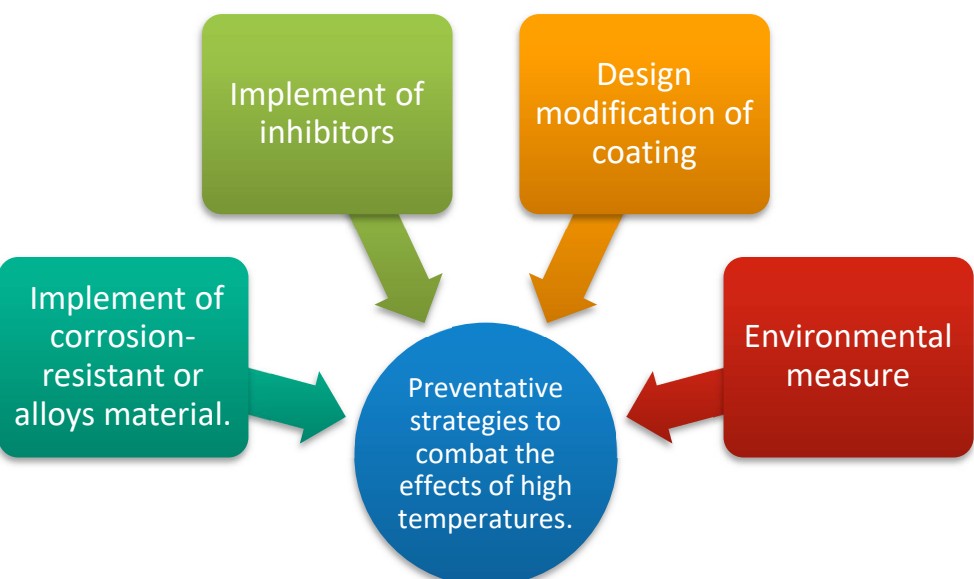

**Figure 21.** Preventative strategies to combat the hot-corrosion effects at high temperatures.

Composite Coating Hot Corrosion Behavior

The composite coating was developed by the researchers for the design modification of coating structures. The 3YSZ-Ni composites have magnetic properties, and for this reason, they can be used as superior surface coating materials and have provided tremendous ability to be used in various working environmental sectors. Solid oxide fuel cell materials (SOFC) can help the ceramic's stability and can be used in the thermal spray deposition process to improve quality surface texture. The thermal barrier coating technique mainly uses high-temperature industrial applications, such as the aerospace industry, to protect the gas turbine components, diesel engine exhaust manifold, and cutting tools application [83–85].

A comprises intra splat fractures, interlamellar pores, and globular pores of various sizes. The porosity of the coating film was calculated using an image analysis procedure from the SEM image (Figure 22). A polished cross-section with $Al_2O_3$/8YSZ TBC coating had an estimated porosity percentage value of around 17%. The $CeO_2$/8YSZ coating porosity was about 10% lower than that of the $Al_2O_3$/8YSZ coating [86].

The 1350 °C as the minimum temperature needed for the ceramic powder grains to achieve the desired rate of compactness on the coated surface structured and sintered effect can also improve the coating microstructure's quality. High densification has been achieved by adding 1.0 mol nickel oxide to a ceramic-based coating material that has provided improved thermal stability in high-temperature thermal shock environmental conditions [87]. A tremendous quantity of research has been conducted on traditional coatings available in previous decades. On the other hand, there are few research types on functionally graded coatings, composite coatings, and bimodal coatings [42,87,88]. Some significant features of this fundamental powder microstructure have provided better corrosion resistance properties [49]. Some studies have represented the number of pores per unit of mass exercise for high importance denser kinetics, influencing microstructural evolution properties during the thermal oxidation test [89]. The Ni/YSZ's shows a great-energy interference by Ni-base powder grains ceramic material over 20 vol.%, which did not exceed the percolation threshold. The electrical conductivity and complicated impedance used for measures corresponded to a nickel concentration of 34 vol.% in TZP

ceramic powder grains [90]. A huge percolation threshold has been associated with one unfinished initial-near order, such as ferrous metal composition within ceramic powder grain materials [91]. The addition of nickel to 3YSZ is transmitted for supreme power to ball milling, and those composites manifest to develop in the densification process. The purified form of nickel elements has represented an anti-prohibitionist tendency into zirconia material because unstable phases and the contact angle may vary from 117° to 122° [92]. Some researchers have proposed a straight reduction of the interfacial reaction among polycrystalline $ZrO_2$ and the liquid phase of Ni, including heating energy, depending on which the interfacial potential at the melting heat for Ni is 1.668 J/m$^2$ [93]. Selection of the nanometric size YSZ powder grains and micro-size of nickel-base powder grain added and then effectively mixing them through a ball milling machine and the highest shrinkage percentage heat for 3YSZ-Ni composites during the process. In this coating process, the two sides' heating improved as the Ni content enhanced the melting temperature, affecting a decreased density with an enhanced Ni amount and a melting temperature rise [94].

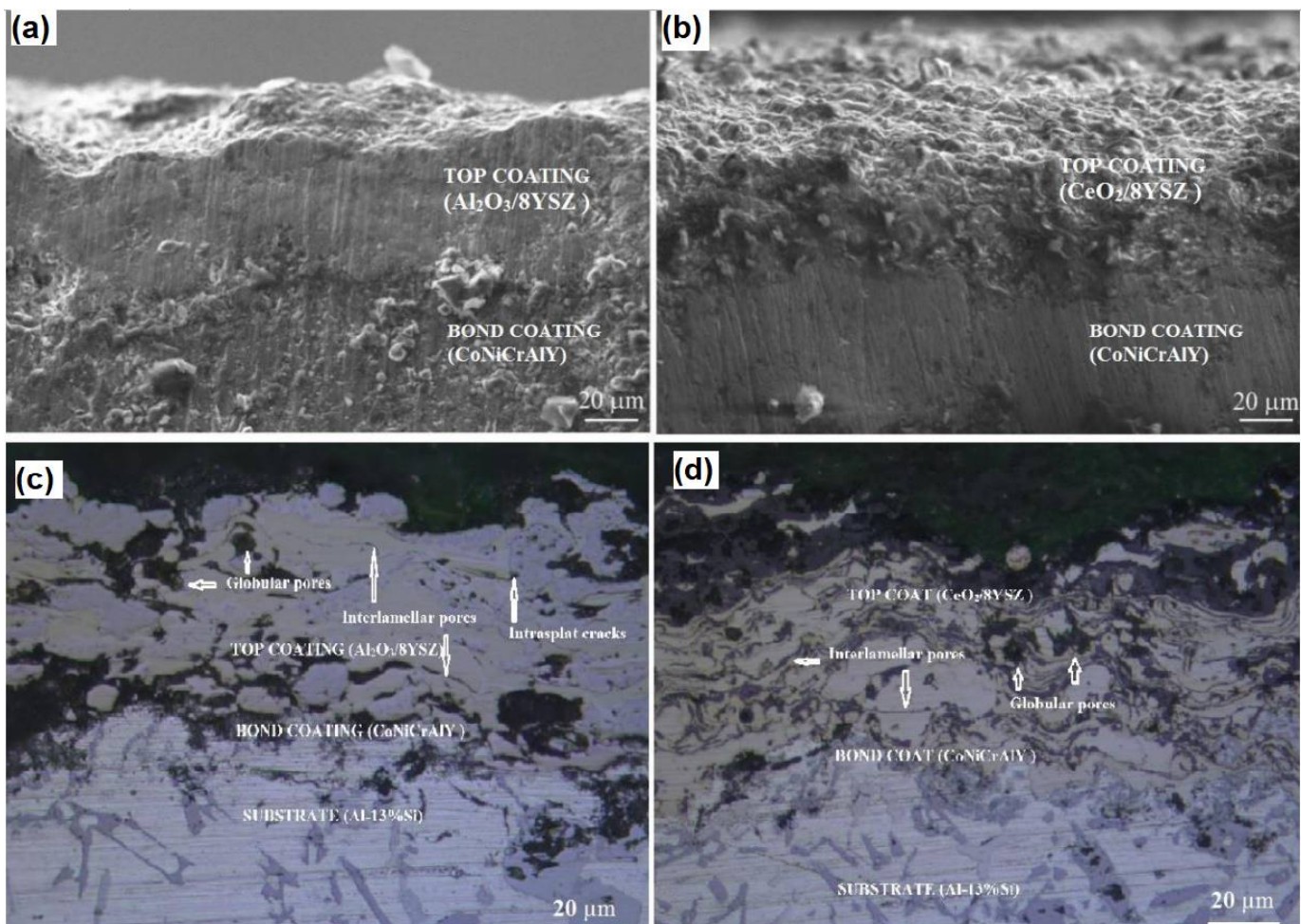

**Figure 22.** Cross-sectional morphologies of as-sprayed TBCs: (**a**) $Al_2O_3$/8YSZ TBCs and (**b**) $CeO_2$/8YSZ TBCs; (**c**) $Al_2O_3$/8YSZ TBCs backscattered election images; (**d**) $CeO_2$/8YSZ TBCs backscattered electron images [86].

Consequently, $CeO_2$/YSZ coating exhibited excellent results by entirely preventing molten salts from flowing in YSZ. Researchers have worked to improve the hot corrosion resistance of YSZ coatings/materials, as shown in Table 2. The molten salts either entirely entered or partially permeated the YSZ covering (Table 2).

**Table 2.** Hot corrosion behavior of TBC-based YSZ coatings in the presence of corrosive salts and infiltration mode.

| Substrate Material | Corrosive Salt Used (wt.%) | Composite Materials | Temperature and Time of Exposure | Technique | Infiltration of Molten Salt in Coating |
|---|---|---|---|---|---|
| Ni-based superalloy | $50 Na_2SO_4 + 50 V_2O_5$ | YSZ [95] | (950 °C, 60 h) | APS | Completely |
| Ni-based superalloy | $NaVO_3$ | YSZ [96] | (900 °C, 300 h) | APS | Completely |
| Ni-based superalloy | $45 Na_2SO_4 + 55 V_2O_5$ | $YSZ + CeO_2$ [97] | (1000 °C, 300 h) | APS | Partially |
| Inconel 738LC superalloy | $55 V_2O_5 + 45 Na_2SO_4$ | $YSZ + CeO_2$ [98] | (1000 °C, 30 h) | APS | Partially |
| Superalloy (Ni3Al) plates | $45 Na_2SO_4 + 55 V_2O_5$ | $8YSZ + Al_2O_3$ [99] | (900 °C, 380 h) | APS | Partially |
| Ni-based superalloy | $NaVO_3$ | $YSZ + CeO_2$ [96] | (900 °C, 300 h) | APS | Partially |
| Ni-based superalloy | $50 Na_2SO_4 + 50 V_2O_5$ | $YSZ/LaMgAl_{11}O_{19}$ [95] | (950 °C, 60 h) | APS | Partially |
| Inconel 738LC superalloy | $50 Na_2SO_4 + 50 V_2O_5$ | $YSZ-Ta_2O_5$ [100] | (1100 °C, 4 h) | Sintering | Partially |
| Nickel superalloy | $45 Na_2SO_4 + 55 V_2O_5$ | $YSZ/La_2Ce_2O_7$ [101] | (910 °C, 30 h) | APS | Partially |
| Inconel 738LC, Ni-15Cr-8.5Co | $45 Na_2SO_4 + 55 V_2O_5$ | $YSZ/Al_2O_3$ [102] | (1050 °C, 40 h) | APS | Partially |
| Inconel 738 LC superalloy | $60 V_2O_5 + 40 Na_2SO_4$ | YSZ [103] | (1000 °C, 100 h) | APS + lasercladding | Completely |
| Inconel 738 | $55 V_2O_5 + 45 Na_2SO_4$ | $YSZ/Al_2O_3$ [104] | (1000 °C, 30 h) | APS + lasercladding | Partially |

### 3.1.2. Thermal Oxidation Performance

The solid form grain particles are striking on the coated materials at very high velocity and high temperature, and then the surface starts to deteriorate. This phenomenon is known as thermal oxidation [105]. The average output of various substrate metals under thermal oxidation environmental conditions was examined in an organized manner, as presented in Table 3. Some significant coating powders have provided better resistance to thermal oxidation, such as $Al_2O_3$, $CrO_3$, $TiO_2$, YSZ, and NI-Cr-based coating. Numerous researchers have proposed decreasing the area of WC particles to the nanostructured to obtain greater power, high-toughness, and wear resistance for WC-Co-Cr coating [106]. Furthermore, the decarburization process's effect occurs in nanostructure coatings. This process has a more comprehensive primary outside dimension by reducing the WC particles' area [107].

**Table 3.** Thermal oxidation performance obtained from different coating materials.

| Year | Substrate Material | Coating Material | Coating Method | Temperature (°C) | Weight Gain of the Coated Substrate (mg/cm$^2$) | Weight Gain of the Uncoated Substrate (mg/cm$^2$) | Percentage Drop after Hot Oxidation |
|---|---|---|---|---|---|---|---|
| 2015 | SC 605 | $Al_2O_3$-40% $TiO_2$ [105] | low-velocity oxy-fuel spray | 800 | 49.12 | 52.79 | 5.64 |
| 2011 | T-22 | Ni-20Cr [108] | Detonation Gun spray | 800 | 16.1 | 46.54 | 65.58 |
| 2006 | T-22 | Ni-20Cr [109] | High-velocity oxy-fuel spray (HVOF) | 900 | 8.26 | 14.99 | 44.98 |
| 2012 | T-91 | Yttria-stabilized zirconia (YSZ) [110] | Plasma spray (PS) | 800 | 8.31 | 69.1 | 73.61 |
| 2014 | ASTM1020 Steel | NiCrC nano [111] | HVOF | 550 | 0.12 | 12.98 | 99.24 |
| | | NiCrC Conventional [111] | HVOF | 550 | 0.29 | 12.99 | 97.69 |
| | | NiCrC nano [111] | HVOF | 650 | 1.1 | 34.1 | 97.07 |
| | | NiCrC Conventional [111] | HVOF | 650 | 1.51 | 34.1 | 95.6 |
| 2014 | INCONEL 625 | $Al_2O_3$-$TiO_2$ [112] | Plasma spray (PS) | 800 | 5 | 15 | 66.68 |
| 2014 | INCONEL 625 | $ZrO_2$ [113] | (PS) | 800 | 1.12 | 15 | 92.68 |
| 2017 | T22 | $CrO_3$ [114] | (PS) | 800 | 50.94 | 65.17 | 21.85 |

### 3.1.3. Recent Approach Use for Improving Hardness and Toughness Values

Complex surface layers have usually been considered to have outstanding wear-resistance characteristics and help for high working temperatures. Several researchers working on the APS coating method find that by applying nanostructured YSZ coating powder to the substrate, metal can provide better coating results than micro-size powder, such as better surface finish and improved abrasion rate. The hardness values of the traditional layer are more meaningful than a nanostructured coating measured by a Vickers indenter test. The use of feedstock in a thermal spray to monitor agglomerated nanostructured powder particles' critical parameters. The embedded most of the nanostructured grains melted after applying a plasma spray jet at a controlled feed rate, which can help form the smooth microstructure on the substrate material. The coated layer did not display any significant nanostructured quality. This layer tended to behave as a conventionalized one.

Nanostructured thermal spray coatings show considerably more excellent resistance to crack propagation than conventional thermal spray coatings. The Vickers indentation technique analyses the creak propagation generated inside the surface coating layer, ensuring its quality and playing a vital role in selecting the proper coating methods. When applied to sufficient weights, the surface coating during Vickers indentation formed cracks and propagated a small distance that edges about the impact of a Vickers indentation. Therefore, the more significant wear of the nanostructured coatings is typically characteristic of excellent protection against crack spread (Table 4).

**Table 4.** Hardness and toughness values obtained from different coating materials in TBC.

| Year | Substrate Metal | Coating Material | Fabrication Process | Hardness Value (HV) | Toughness (MPa$\sqrt{m}$) |
|---|---|---|---|---|---|
| 2008 | MS | $ZrB_2$–SiC/$ZrO_2$ [115] | Suspension plasma spraying (SPS) | 20–18 GPa | |
| 2003 | MS | $Al_2O_3$–$Ti_3SiC_2$ [116] | (SPS) | 4–17 GPa | |
| 2013 | SS-316 | TiB–Ti [117] | (SPS) | 5.8–17 GPa | |
| 2012 | SS-316 | HAp–$Al_2O_3$–YSZ [107] | (SPS) | 6–13.9 GPa | |
| 2014 | IN-738LC | W–Cu [118] | (SPS) | 4–5 GPa | 1.4 ± 0.11 |
| 2009 | IN-800LC | SiC–$Al_3BC_3$ [119] | (SPS) | 18.5–26.4 GPa | 1.6 ± 0.09 |
| 2013 | IN-738LC | WC–TiC–$Cr_3C_2$ [120] | (SPS) | 18.4–23.2 GPa | 0.9 ± 0.12 |
| 1999 | SS-304 | NiCrAl-$MgZrO_3$ [121] | Plasma spray (PS) | 900–350 BHN | |
| 2007 | IN-625LC | $ZrO_2$–$Al_2O_3$ [122] | (PS) | 1170–870 | |
| 2003 | SS-304 | NiCrAl/$MgZrO_3$ [123] | (PS) | 150–220 | |
| 2008 | AISI-410 | $TiO_2$–HAp [124] | (PS) | 363.9–513.7 | |
| 2019 | CA6NM | Ni-40$TiO_2$ [125] | (HVFS) | 605 ± 37 | 1.8 ± 0.2 |
| | | Ni-20$TiO_2$ + 20$Al_2O_3$ [125] | (HVFS) | 585 ± 33 | 1.6 ± 0.2 |
| 2016 | SS-304 | WC-10Co-4Cr nanostructure [106] | HVOF sprayed | 1696 ± 46 | 2.01 ± 0.04 |
| | | WC-10Co-4Cr conventional [106] | HVOF sprayed | 1147 ± 50 | 3.61 ± 0.31 |
| 2020 | SS-304 | MWCNTs reinforced with nano-WC-Co-Cr [126] | HVOF sprayed | 1530 ± 143 | 9.14 ± 1.30 |
| 2012 | 13Cr–4Ni | Colmonoy 88 and stellite-6 [127] | Laser surface modification | 674–800 | |
| 2006 | MS | Ni-Cr-Si-B [128] | Flame sprayed | 280 | |
| 2010 | 21Cr–4Ni–N steel | stellite-6 [129] | D-Gun | 1098 | |
| | | $Cr_3C_2$-NiCr [129] | D-Gun | 824 | |
| | | WC-Co-Cr [129] | D-Gun | 990 | |
| 2011 | MS | Inconel-718 [130] | APS process | 340 | |
| 2012 | 13Cr$_4$Ni and 16Cr$_5$Ni steels | WC-Co-Cr on 13Cr$_4$Ni [131] | D-Gun | 1401 | |
| | | WC-Co-Cr on 16Cr$_5$Ni [131] | D-Gun | 1422 | |
| 2012 | CF8M | WC-10Co-4Cr [132] | D-Gun | 1160 | |
| | | $Al_2O_3$ + 13$TiO_2$ [132] | D-Gun | 830 | |
| 2012 | CF8M and CA6NM | $Cr_2O_3$ on CF8M [133] | HVOF sprayed | 1200 | |
| | | $Cr_2O_3$ on CA6NM [133] | | 1400 | |
| 2012 | CF8M | WC-10Co-4Cr [134] | D-Gun | 1120 | |
| 2012 | Ni based superalloy | $ZrO_2$–8 mass-%$Y_2O_3$ [135] | (APS) | 72 GPa | 2.0 ± 0.3 |
| 2016 | IN-718LC | YSZ ($ZrO_2$ + $Y_2O_3$) [136] | (APS) | 680 | |
| | IN-718LC | YSZ ($ZrO_2$ + $Y_2O_3$) [136] | CGDS | 740 | |
| 2014 | IN-718LC | 8YSZ [137] | (APS) | 58.04 GPa | 1.58 ± 0.15 |
| 2020 | Stainless steel | WC-10Co-4Cr [138] | TIG welding | 713.13–879.51 | 3.84 ± 0.11 |
| 2017 | IN-738LC | $Al_2O_3$/CSZ [139] | (APS) | 172.05 GPa | |
| 2019 | 35CrMo | WC-10Co4Cr [140] | HVOF | 1316 ± 45 | 7.11 ± 0.10 |
| 2014 | AISI 1045 steel | WC–10Co–4Cr [141] | HVOF | 983 | |
| 2018 | CA6NM steel | WC-10Co-4Cr [142] | High-Velocity Oxy Liquid Fuel (HVOLF) | 1402 ± 16 | |
| 2011 | Stainless steel | Ni [143] | Electrodeposition | 320 ± 15 | |
| | | Ni–CNT [143] | Electrodeposition | 580 ± 15 | |
| 2017 | Pure Brass | Cu–MWCNTs [144] | Electrodeposition | 335 | |
| 2014 | SAE 1045 steel | WC–10Co–4Cr [145] | HVOF | 11.1 GPa | |
| | | WC–12Co [145] | HVOF | 9.3 GPa | |
| 2012 | SAE 1045 steel | WC–10Co–4C [146] | HVOF | 261 ± 10 | 1.42 ± 0.80 |
| | | WC–12Co [146] | HVOF | 254 ± 12 | 4.56 ± 0.90 |
| 2017 | IN-738LC | 8YSZ conventional [147] | (APS) | 421 | |
| | | 8YSZ nanostructured [147] | (APS) | 327 | |

The Vickers-indentation morphology of nanostructure and traditional 8YSZ coatings where the load applied is 5 kgf (49 N) is shown in Figure 23. Figure 23 also shows that those produced cracks are thinner and shorter inside nanostructure 8YSZ coatings, while the generated cracks are coarser and longer in traditional 8YSZ layers [147]. Estimated and presented a few tests to show partially melted nanostructured powder grains deposited on that coated morphology. A spray jet tends to work mainly with crack arresters, thus improving coating toughness [148].

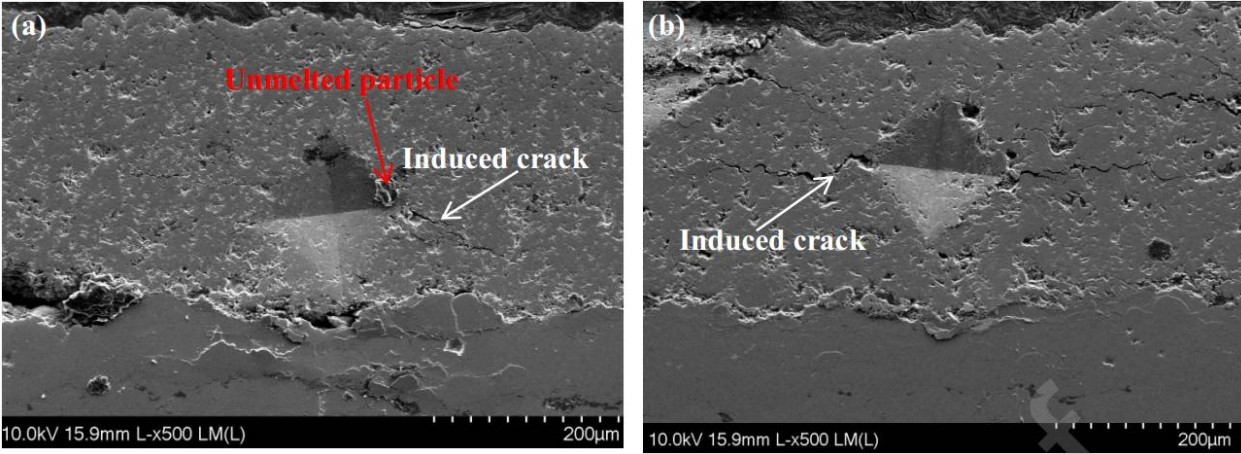

**Figure 23.** Morphology of indentation of (**a**) nanostructure and (**b**) traditional ceramic coatings after application (5 kgf) [148].

The Effect on Energy Efficiency of Yttria-Stabilized Zirconia non Brass, Copper and Hardened Steel Nozzle in Additive Manufacturing was analyzed, which is the recent advancement of the application of TBCs in additive manufacturing [149–152]. Other methods can also be used, such as electrochemical deposition regimes, to develop thin films [153,154]. Nano-composite materials can be used to counter hot corrosion in cases of high-temperature applications [33,155]. The YSZ coating system is going through a phase, where advancement in the composition is taking place [156] and the main process used for the deposition of ceramics is plasma spraying technique [157–159].

## 4. Perspectives and Summary

This research study's primary focus is to analyze and summarize technical advancements concerning the essential features of TBCs, with stress on the correlation between the thermal, mechanical and erosion properties of the coating material by selecting the suitable coating method. In this research study, many efforts in this area aim to improve the microscopic-level properties of TBC materials that increase their durability while working in high-temperature and highly corrosive conditions to prepare them to fit for various industrial applications. This review article contains the introductory section and some other sections.

1.   It discusses the context and applications of TBCs, the elements of a traditional TBC framework and their function and required properties.
2.   A TBC system's output is closely related to the methods used in its development. Different processing techniques and advancements in material depositing methods have versatile tailoring of their microstructure and combinations for special engineering applications.
3.   Approaches used to TBC design and coating materials. Develop TBC; the different design methodologies are implemented to address the coating's working specifications and stability. Those involve functionally graded coating, composite coating, and multi-layered coating arrangements, which are also discussed in this review.

4. Some challenges of TBC include improving performance during testing thermal oxidation, mechanical properties, such as hardness and toughness, and hot corrosion obtained from different coating materials and other material deposition techniques.

5. The recommendations for thermal barrier coatings include the use of electron beam-physical vapor deposition (EB-PVD) or plasma spray. Both of these processes can best rain compliant, have low thermal conductivity, and generate erosion-resistant coatings whose microstructures can be tailored for the desired application. Moreover, the composition of the top coat can vary with the addition of other phases.

**Author Contributions:** A.M., Original draft preparation; H.V., Supervisor, review and editing; S.S., supervisor; C.P., Conceptualization and visualization; K.K.S., Conceptualization; E.L., Formal analysis and funding; D.B., Formal analysis and software; J.X., Analysis. All authors have read and agreed to the published version of the manuscript.

**Funding:** This research was funded by a grant from the Romanian Ministry of Research, Innovation and Digitalization, project number PFE 26/30.12.2021, PERFORM-CDI@UPT100—The increasing of the performance of the Polytechnic University of Timișoara by strengthening the research, development and technological transfer capacity in the field of "Energy, Environment and Climate Change" at the beginning of the second century of its existence, within Program 1—Development of the national system of Research and Development, Subprogram 1.2—Institutional Performance—Institutional Development Projects—Excellence Funding Projects in RDI, PNCDI III.

**Institutional Review Board Statement:** Not applicable.

**Informed Consent Statement:** Not applicable.

**Data Availability Statement:** The authors did not report any data other than manuscript.

**Conflicts of Interest:** The authors declare no conflict of interest.

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
