# Peer review of "Processing and Advancements in the Development of Thermal Barrier Coatings: A Review"

_coatings, doi:10.3390/coatings12091318_

Round 1

Reviewer 1 Report

Mehta et al present a good paper entitled as Processing and advancements in the development of thermal 2 barrier coatings: A review. I would recommend this paper suitable for publication in this journal coating after some minor concerns 

1. Abstract need to be revised and grammar and tropical error need to be corrected 

2. Introduction need to be cited with latest references regarding your study such as 10.1016/j.mattod.2022.04.002

3. Author should explain why their is need of this review?

4. Scope of review should be explained in details

5. Author should explain a comparitive no of papers published by scoupus on this topic

6. Author should explain future prospective and recommendations in conclusion 

Author Response

REVIEWER 1

Mehta et al present a good paper entitled as Processing and advancements in the development of thermal 2 barrier coatings: A review. I would recommend this paper suitable for publication in this journal coating after some minor concerns

  1. Abstract need to be revised and grammar and tropical error need to be corrected

The authors have revised the abstract and checked for tropical error in the revised paper

  1. Introduction need to be cited with latest references regarding your study such as 10.1016/j.mattod.2022.04.002

The reference has been cited in the introduction part of revised manuscript.

  1. Author should explain why there is need of this review?

TBCs are being used to insulate gas turbine components that are subjected to excessive temperatures. Since low thermal conductivity allows these coatings to act as thermal barriers, the metal surface temperatures are decreased. Lower metal temperatures are given greater component durability by reducing creep stress and fatigue, while also reducing oxidation and corrosion rates. The advancements are going on in the field of high temperature applications. The materials that can sustain high temperature needs to be optimized. The thermal barrier coatings need optimization in terms of resisting the high temperature oxidation, erosion and hot corrosion.  This paper emphasis on the recent techniques used in the development of thermal barrier coatings and their performance has been analysed. The authors have incorporated the need of this review article in the revised manuscript.

The authors have incorporated the need of this review as a novelty part at the end of the introduction section.

  1. Scope of review should be explained in details

The authors have incorporated the scope of the review in the revised manuscript.

Approaches use to TBC design and coating materials. Development of TBCs; the different design methodologies are implementing to address the coating's working specifications and stability. Those involve functionally graded coating, composite coating, and multi-layered coating arrangements also discussed in this review.

  1. Author should explain a comparitive no of papers published by scoupus on this topic

The above mentioned data has been provided in the revised manuscript.

  1. Author should explain future prospective and recommendations in conclusion

The future prospective and recommendations have been added in the conclusion part of the revised manuscript.

Reviewer 2 Report

The authors mentioned that TBC is used in gas turbine blades and diesel engines. The TBC method has many other applications. Adding these will strengthen the scientific aspect of the article. As an example, the articles given below definitely have to add to the references and it should be mentioned that TBC technology can also be used in additive manufacturing. 1) DOI: 10.3390/coatings12050690 , 2) DOI: 10.3390/coatings11070792 .

Author Response

REVIEWER 2

The authors mentioned that TBC is used in gas turbine blades and diesel engines. The TBC method has many other applications. Adding these will strengthen the scientific aspect of the article. As an example, the articles given below definitely have to add to the references and it should be mentioned that TBC technology can also be used in additive manufacturing. 1) DOI: 10.3390/coatings12050690  2) DOI: 10.3390/coatings11070792 .

Thanks for your valuable suggestions. The authors have incorporated the suggestions given by the reviewer in the revised manuscript.

Reviewer 3 Report

Referee Report

on paper “ Processing and advancements in the development of thermal barrier coatings: A review â€œ (coatings-1892152) by author Amrinder Mehta, Hitesh Vasudev, Sharanjit Singh, Chander Prakash, Kuldeep K. Saxena, Emanoil Linul, Dharam Buddhi, Jinyang Xu submitted to Coatings

This is interesting review paper. It reports provides a brief overview of the various coating methods and design methodologies used for the thermal barrier coatings to improve the coating's surface quality. This review's primary focus is on the growth, preparation, and properties of developed thermal barrier coatings that have contributed towards specific surface coatings for industrial applications. The presented results are reliable without any doubts. However, I have some comments and additions. I would like to note a few points to improve the paper before it can be published:

1.   It is necessary to give an idea of additional methods for the formation of films and coatings to improve the surface quality of the material:

(1). D.I. Tishkevich, S.S. Grabchikov, L.S. Tsybulskaya, V.S. Shendyukov, S.S. Perevoznikov, S.V. Trukhanov, E.L. Trukhanova, A.V. Trukhanov, D.A. Vinnik, Electrochemical deposition regimes and critical influence of organic additives on the structure of Bi films, J. Alloys Compd. 735 (2018) 1943-1948. https://doi.org/10.1016/j.jallcom.2017.11.329.

(2). D.I. Tishkevich, S.S. Grabchikov, S.B. Lastovskii, S.V. Trukhanov, T.I. Zubar, D.S. Vasin, A.V. Trukhanov, Correlation of the synthesis conditions and microstructure for Bi-based electron shields production, J. Alloys Compd. 749 (2018) 1036-1042. https://doi.org/10.1016/j.jallcom.2018.03.288.

2.   It is necessary to give information about composite materials are perspective for practical applications:

(3). M.A. El-Ghobashy, H. Hashim, M.A. Darwish, M.U. Khandaker, A. Sulieman, N. Tamam, S.V. Trukhanov, A.V. Trukhanov, M.A. Salem, Eco-friendly NiO/polydopamine nanocomposite for efficient removal of dyes from wastewater, Nanomaterials 12 (2022) 1103. https://doi.org/10.3390/nano12071103.

(4). D.I. Tishkevich, T.I. Zubar, A.L. Zhaludkevich, I.U. Razanau, T.N. Vershinina, A.A. Bondaruk, E.K. Zheleznova, M. Dong, M.Y. Hanfi, M.I. Sayyed, M.V. Silibin, S.V. Trukhanov, A.V. Trukhanov, Isostatic hot pressed W-Cu composites with nanosized grain boundaries: microstructure, structure, and radiation shielding efficiency against gamma-rays, Nanomaterials 12 (2022) 1642. https://doi.org/10.3390/nano12101642.

3.   The proposed 4 papers should be inserted in References.

The paper should be sent to me for the second analysis after the moderate revisions.

Author Response

REVIEWER 3

 “ Processing and advancements in the development of thermal barrier coatings: A review “ (coatings-1892152) by author Amrinder Mehta, Hitesh Vasudev, Sharanjit Singh, Chander Prakash, Kuldeep K. Saxena, EmanoilLinul, Dharam Buddhi, Jinyang Xu submitted to Coatings

This is interesting review paper. It reports provides a brief overview of the various coating methods and design methodologies used for the thermal barrier coatings to improve the coating's surface quality. This review's primary focus is on the growth, preparation, and properties of developed thermal barrier coatings that have contributed towards specific surface coatings for industrial applications. The presented results are reliable without any doubts. However, I have some comments and additions. I would like to note a few points to improve the paper before it can be published:

  1. It is necessary to give an idea of additional methods for the formation of films and coatings to improve the surface quality of the material:

(1). D.I. Tishkevich, S.S. Grabchikov, L.S. Tsybulskaya, V.S. Shendyukov, S.S. Perevoznikov, S.V. Trukhanov, E.L. Trukhanova, A.V. Trukhanov, D.A. Vinnik, Electrochemical deposition regimes and critical influence of organic additives on the structure of Bi films, J. Alloys Compd. 735 (2018) 1943-1948. https://doi.org/10.1016/j.jallcom.2017.11.329.

(2). D.I. Tishkevich, S.S. Grabchikov, S.B. Lastovskii, S.V. Trukhanov, T.I. Zubar, D.S. Vasin, A.V. Trukhanov, Correlation of the synthesis conditions and microstructure for Bi-based electron shields production, J. Alloys Compd. 749 (2018) 1036-1042. https://doi.org/10.1016/j.jallcom.2018.03.288.

Thanks for your valuable suggestions. The authors have incorporated the suggestions given by the reviewer in the revised manuscript.

  1. It is necessary to give information about composite materials are perspective for practical applications:

(3). M.A. El-Ghobashy, H. Hashim, M.A. Darwish, M.U. Khandaker, A. Sulieman, N. Tamam, S.V. Trukhanov, A.V. Trukhanov, M.A. Salem, Eco-friendly NiO/polydopamine nanocomposite for efficient removal of dyes from wastewater, Nanomaterials 12 (2022) 1103. https://doi.org/10.3390/nano12071103.

(4). D.I. Tishkevich, T.I. Zubar, A.L. Zhaludkevich, I.U. Razanau, T.N. Vershinina, A.A. Bondaruk, E.K. Zheleznova, M. Dong, M.Y. Hanfi, M.I. Sayyed, M.V. Silibin, S.V. Trukhanov, A.V. Trukhanov, Isostatic hot pressed W-Cu composites with nanosized grain boundaries: microstructure, structure, and radiation shielding efficiency against gamma-rays, Nanomaterials 12 (2022) 1642. https://doi.org/10.3390/nano12101642.

Thanks for your valuable suggestions. The authors have incorporated the suggestions given by the reviewer in the revised manuscript.

  1. The proposed 4 papers should be inserted in References.

Thanks for your valuable suggestions. The authors have incorporated the suggestions given by the reviewer in the revised manuscript.

Round 2

Reviewer 2 Report

An indication of the scientific value of the article is the references used. Coatings crosschecks references. For this reason, it is important to find the doi numbers of the references and to write the reference information correctly. For example, author names have to be [Demir H, Cosgun A.E.C.] in reference (149 and 152) and [Demir H.] in reference 150.

The article contains valuable scientific information. After making the necessary corrections, I believe that the article will be a good reference for future studies.

Author Response

An indication of the scientific value of the article is the references used. Coatings crosschecks references. For this reason, it is important to find the doi numbers of the references and to write the reference information correctly. For example, author names have to be [Demir H, Cosgun A.E.C.] in reference (149 and 152) and [Demir H.] in reference 150.

The article contains valuable scientific information. After making the necessary corrections, I believe that the article will be a good reference for future studies.

The same has been updated in the revised manuscript.

Reviewer 3 Report

Referee Report

on paper “ Processing and advancements in the development of thermal barrier coatings: A review “ (coatings-1892152-v2) by author Amrinder Mehta, Hitesh Vasudev, Sharanjit Singh, Chander Prakash, Kuldeep K. Saxena, Emanoil Linul, Dharam Buddhi, Jinyang Xu submitted to Coatings

This paper has been well corrected and it should be published.

Author Response

on paper “ Processing and advancements in the development of thermal barrier coatings: A review “ (coatings-1892152-v2) by author Amrinder Mehta, Hitesh Vasudev, Sharanjit Singh, Chander Prakash, Kuldeep K. Saxena, Emanoil Linul, Dharam Buddhi, Jinyang Xu submitted to Coatings 

This paper has been well corrected and it should be published.

Thanks for accepting our work.